# A Novel Security Threat Model for Automated AI Accelerator Generation Platforms

## Abstract

In recent years, the design of Artificial Intelligence (AI) accelerators has gradually shifted from focusing solely on standalone accelerator hardware to considering the entire system, giving rise to a new AI accelerator design paradigm that emphasizes full-stack integration. Systems designed based on this paradigm offer a user-friendly, end-to-end solution for deploying pre-trained models. While previous studies have identified vulnerabilities in individual hardware components or models, the security of this paradigm has not yet been thoroughly evaluated. This work, from an attacker's perspective, proposes a threat model based on this paradigm and reveals the potential security vulnerabilities of systems by embedding malicious code in the design flow, highlighting the necessity for protection to address this security gap. In exploration and generation, maliciously leverage the exploration unit to identify sensitive parameters in the model's intermediate layers and insert hardware Trojan (HT) into the accelerator. In execution, malicious information is concealed within the control instructions, triggering the HT. Experimental results demonstrate that the proposed method, which manipulates sensitive parameters in a few selected kernels across the middle convolutional layers, successfully misclassifies input images into specified categories with high misclassification rates across various models: 97.3% in YOLOv8 by modifying only three parameters per layer in three layers, 99.2% in ResNet-18 by altering four parameters per layer in three layers and 98.1% for VGG-16 by changing seven parameters per layer in four layers. Additionally, the area overhead introduced by the proposed HT occupies no more than 0.34% of the total design while maintaining near-original performance as in uncompromised designs, which clearly illustrates the concealment of the proposed security threat.

## 1 Introduction

Recent research has revealed that AI accelerator are inherently vulnerable to various attacks, highlighting security as a significant concern. These vulnerabilities stem from multiple sources: **1)** The models themselves, which allows attackers to induce malfunction through minor perturbations to the inputs (Baniecki & Biecek, 2024; Costa et al., 2024). Besides adding perturbations to the inputs, attackers can also control the model's behavior by attacking the model parameters (Rakin et al., 2019; 2021; Bai et al., 2023). **2)** The accelerator hardware architecture, such as memory, is susceptible to attacks such as the Row Hammer Attack (Kim et al., 2014). Attackers can also exploit timing violations to cause the entire design to malfunction (Liu et al., 2020b; Mukherjee & Chakraborty, 2022). **3)** Security vulnerabilities during the design synthesis process, such as attackers tampering with the Look-Up Tables (LUTs) in Field-Programmable Gate Array (FPGA) (Nozaki et al., 2020; Krieg et al., 2016).

Lately, the research focus on AI accelerators has gradually shifted from solely focusing on AI accelerators to encompassing the entire System on Chip (SoC) and its related software stack, giving rise to a new AI accelerator design paradigm (Xilinx, 2024; Genc et al., 2021). The yellow part in Fig. 1 provides a sketch of this paradigm, which centers not only on optimizing the performance and efficiency of accelerators but also on the deep co-design between hardware and software, as well as their integration at the system level to address increasingly complex AI workloads. Key components of this paradigm include exploration unit (e.g., Design Space Exploration (DSE)) and middleware. DSE uses exploration algorithms to evaluate various design parameters to optimize

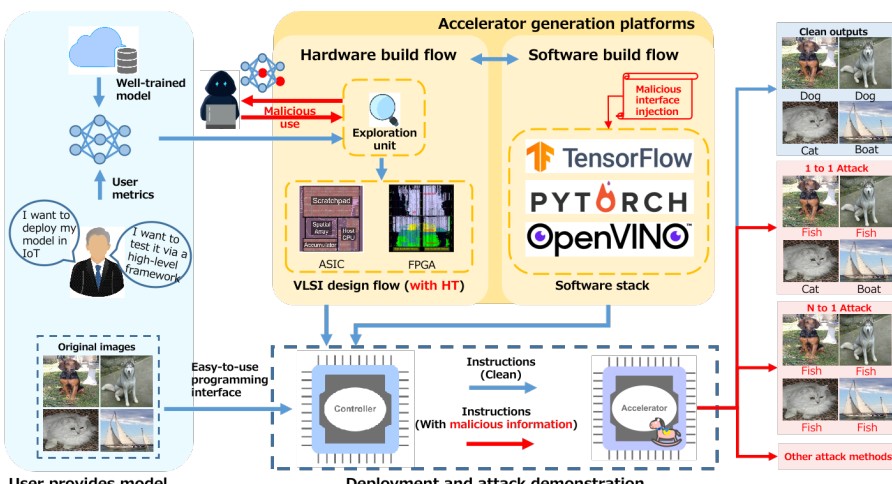

Figure 1: Overview of the AI accelerator design paradigm: the left blue part shows the user-provided model and requirements, the middle yellow part represents the accelerator generation platform mapping the model to hardware with a user-friendly interface, and the right blue part provides application examples. The attacker's target is the red part within the accelerator generation platform.

hardware designs under resource constraints, ensuring the system achieves the best balance between performance, power consumption, and area (Chen et al., 2024; Cai et al., 2022). For instance, in accelerators based on pipeline architecture, DSE can allocate appropriate processing elements (PEs) for different models to ensure latency balance across layers and maximize system throughput (Zhang et al., 2018). Meanwhile, middleware provides an abstraction layer that simplifies the interaction between developers and accelerators, hiding the complexity of the underlying hardware and enhancing the usability and scalability of the system. Significant progress has been made in areas such as hardware and software co-design and system integration within the AI accelerator design paradigm. More specifically, these platforms have recently attracted increasing attention due to their low-cost realization, quick implementation, and easy deployment on IoT devices and low-cost edge devices. Unfortunately, the security aspects of these black-box-like platforms remain largely unexplored.

This research aims to propose a generic threat model targeting the AI accelerator design paradigm. Under this model, the system analyzes user-provided pre-trained models to generate the optimal hardware and software stack. Concurrently, an attacker exploits malicious code embedded in the hardware generation flow to analyze model vulnerabilities and insert a hardware Trojan (HT). In parallel, the software generation flow is compromised, allowing the injection of malicious instructions into the communication protocol between the controller and the accelerator, effectively triggering the HT. The code for this research is available at https://github.com/AnonymousCode-HT/C-SFE.git.

The main contributions are as follows: **1)** A general threat model for automated AI accelerator generation platforms is proposed. It explores how attackers can integrate malicious code into platform components to analyze model vulnerabilities and then insert HTs that can be triggered via a middleware-based method to achieve desired outcomes with negligible area overhead and no performance degradation. **2)** A Cross-layer Sensitive Filter Exploration (C-SFE) targeted attack algorithm is presented and embedded in the hardware design flow. C-SFE is a bit-level adversarial weight attack, differing from previous research by using a heuristic algorithm and relying solely on forward propagation without gradient information. It targets intermediate convolutional layers, and parameters identified by C-SFE exhibit regularity, making it more compatible with HT design. **3)** The proposed threat model was validated on the state-of-the-art automated AI accelerator generation platform, and the actual attack effectiveness on the YOLOv8 classification model, ResNet-18, and VGG-16 was tested on FPGA.

## 2 PRELIMINARIES

**Hardware Trojans** are malicious circuits that are secretly inserted into a circuit. In general, they remain silent and are triggered only at certain moments, causing a serious impact on the functionality

of the circuit. The hardware Trojan usually consists of a trigger and a payload, during normal circuit operation, the trigger will monitor a certain (or some) signal, when these signals reach a specific condition, it will control the payload into working state, the payload is responsible for the specific attack. Many different types of hardware Trojans can be created based on physical, activation, and behavioral characteristics (Xue et al., 2020).

**AI accelerator design paradigm** focuses on optimizing the performance, energy efficiency, and scalability of deep neural network workloads through specialized hardware architectures. It emphasizes the co-design of hardware and software to support a wide range of applications, from high-performance computing to low-power edge devices, leading to a significant number of tunable parameters. As a result, exploration unit within the paradigm, such as DSE, become essential for optimizing accelerator performance and energy efficiency, helping designers choose the best configurations. Lately, many researchers have explored this design paradigm, with notable examples such as Gemmini (Genc et al., 2021) and NVDLA (Zhou et al., 2018). Gemmini is a full-stack DNN accelerator generator featuring: **1)** Flexible hardware parameterization supporting various dataflows (e.g., weight stationary, output stationary). **2)** Integration with the Chipyard framework (Amid et al., 2020), enabling tight coupling with RISC-V Rocket cores via the Rocket Custom Coprocessor (RoCC) commands. **3)** A multi-layer software stack based on ONNX Runtime (Microsoft, 2024), providing an easy-to-use interface for end users and low-level control via C/C++ for system programmers. Some researchers have also integrated NVDLA into the Chipyard framework to provide a unified approach to accelerator generation (Gonzalez & Hong, 2020; Farshchi et al., 2019). However, inconsistencies in communication interfaces remain. For example, Gemmini uses RoCC commands to communicate with RISC-V Rocket cores, while NVDLA relies on Memory-Mapped I/O (MMIO). The optimization and exploration algorithms in these automatic accelerator generation platforms might be maliciously used for HT insertion; however, this security risk has not been investigated to date.

## 3 ATTACK FRAMEWORK

### 3.1 THREAT MODEL

As shown in Fig. 1, the proposed method belongs to the category of grey-box attacks. It is assumed that the adversary is the developer of the AI accelerator platform, and the platform's level of openness is similar to Vitis AI (Xilinx, 2024). This means that due to commercial purposes, its internal exploration unit such as DSE are not made public. Therefore, the platform is transparent to the user, and the design output from the 'VLSI design flow' is also transparent. However, the user can control the accelerator's behavior through the controller's instructions, such as reading data from a specified location in memory into the accelerator and performing matrix multiplication or convolution operations.

During the design and generation phase, since the platform receives information about the model structure and parameters (i.e., 'User provides model' in Fig. 1), the adversary would also have this information. Additionally, the adversary cannot access the training set but can access a small dataset for validation purposes for two key reasons. First, because floating-point data is inefficient in terms of both computational performance and hardware resource utilization, quantization methods are extensively employed during the design space exploration process. In our approach, we employ the Post-Training Quantization (PTQ) method (included in the 'Exploration unit'), which in turn necessitates the use of a compact validation dataset. Second, after the quantization process, the quantized model requires a small subset of input data to determine if there is a significant difference in model accuracy compared to the floating-point scenario.

### 3.2 ATTACK PROCEDURE

As shown in the red part of Fig. 1, in this threat model, the attacker exploits three components in the generation platform that lack sufficient security focus: **1)** Exploration unit, such as DSE, typically integrates various exploration algorithms to find system parameter settings that best meet user requirements. For example, SkyNet uses a group-based particle swarm optimization (PSO) algorithm to evolve network candidates for higher accuracy and efficiency (Zhang et al., 2020). Attackers can insert malicious exploration algorithms into it, or, to reduce the size of the malicious

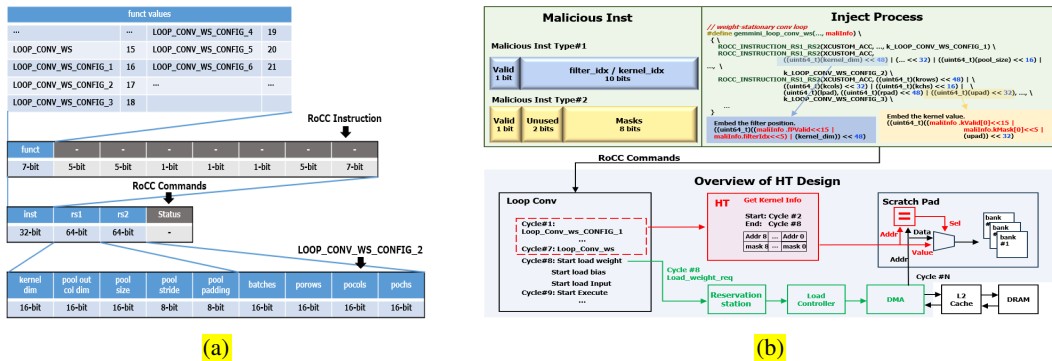

Figure 2: (a) The structure of RoCC commands; (b) Overview of the attack process, with the integration of malicious information in the upper part and the HT within the accelerator in the lower part.

code, they can reuse the exploration algorithms within the DSE to explore the target parameter values. **2)** Software stack, which provides control interfaces for system developers or end users to interact with the accelerator. Since it is ultimately compiled into dynamic libraries like .so files for developers, or Python wheels for end users, it is difficult for them to detect any hidden interfaces or subtle code modifications embedded within. **3)** Hardware generation, such as RTL code or IP generation, can be exploited to insert HT, allowing the attacker to monitor instructions sent by the controller (via the software stack) and trigger an attack when specific information is detected.

The workflow and attack process are as follows (see Appendix A.1 for a simplified representation): **1)** The user intends to deploy their trained model onto domain-specific hardware (e.g., edge devices focusing on low-power applications). Therefore, they provide the trained model, calibration input images, and requirements (e.g., energy-efficiency priority). **2)** The Exploration unit configures the optimal design parameters based on the user's requirements. Simultaneously, the embedded malicious code leverages the exploration algorithm to locate target model parameters (e.g., parameters for executing an N-to-1 attack). **3)** Using the optimal design parameters configured by the Exploration unit in the previous step, the VLSI design flow automatically generates the RTL code for the SoC (including the controller and accelerator), inserts the HT into the accelerator, and then produces a bitfile (for FPGA) or layout (for ASIC) based on user requirements for delivery. **4)** The software stack is adapted to the generated design, enabling the user to easily control the accelerator. During the generation process, malicious code is inserted so that when the control instructions are sent to the accelerator, they carry the information of the target parameters to be attacked.

There are various methods to trigger the attack, such as a timing trigger (i.e., the malicious code embedded in the software stack activates at a specific interval, such as once a week), a random trigger (i.e., activation occurs when a generated random number matches a predefined value), or a targeted trigger (i.e., activation after continuously detecting a specific category). As shown on the right side of Fig. 1, once the HT embedded within the accelerator is triggered, it executes attacks such as a 1-to-1 attack, where a specific class is misclassified into an attacker-specified class, or an N-to-1 attack, where all classes are misclassified into the attacker-specified class. The information necessary for the attack is concealed within control instructions, and the specific effects of the attack are determined by the malicious algorithm embedded in the exploration unit. In the following section, we take Gemmini as an example to demonstrate how this threat model operates in a real AI accelerator generation platform. Notably, our approach is broadly applicable to any similar automation platform.

### 3.3 ATTACK EXAMPLE: THE GEMMINI CASE

Gemmini leverages Complex Instruction Set Computer (CISC) instructions to simplify accelerator operations, such as performing multi-step convolution configurations with a single instruction, thereby reducing command transmissions and improving throughput. These CISC instructions are embedded in the ONNX Runtime middleware, which automatically allocates computations to Gemmini when users input models in ONNX format. Since the middleware operates on the Rocket Core, the CISC instructions are ultimately translated into RoCC commands for execution by Gemmini.

**RoCC commands**. Fig. 2a illustrates the structure of RoCC commands, where the `funct` field specifies the accelerator operation, the `rs1` and `rs2` fields contain the necessary information. For example, the configuration command `LOOP_CONV_WS_CONFIG_2`, which is part of the CISC instruction for the convolution operation, includes information such as kernel dimension and pool size.

**Overview of the Attack Process**. Gemmini provides a dedicated CISC instruction called `gemmini_loop_conv_ws` to improve throughput for convolution operations which require transmitting six consecutive configuration commands followed by the execution command (i.e., `LOOP_CONV_WS_CONFIG_1` to `LOOP_CONV_WS_CONFIG_6` and `LOOP_CONV_WS` in Fig. 2a). Notably, fields such as `pool size` and `kernel dim` occupy 16 bits in the instruction, although in practice they do not require such a wide bit width. For example, in ResNet-18, the largest convolution kernel dim is 7, and the largest pool size is 3, using only 3 and 2 bits out of the 16 bits, respectively. Therefore, the unused bits in these fields can be exploited by attackers to embed malicious information. Based on this, 15 similar fields were identified among the seven RoCC commands mentioned above. Of these, 5 bits are reserved for original information, while the remaining 11 bits transmit malicious instructions, with their structure illustrated in the 'Malicious Inst' section of Fig. 2b. The malicious instructions are categorized into two types: the first type transmits location information using one valid bit and ten data bits, allowing the indexing of up to 1024 positions. The second type transmits the mask for the attacked parameters, with the lower 8 bits specifying which bits to flip. Therefore, to attack a $3 \times 3$ kernel within a convolutional layer, two fields are needed to specify the filter's [1] position and the kernel's offset, and if all 9 parameters are targeted, a total of 11 fields will be required, which is within the available limit of 15 fields. As shown in the lower part of Fig. 2b, the `LoopConv` module of Gemmini receives one instruction per cycle, taking seven cycles in total. From the eighth cycle, the module signals the `Load Controller` to load weight parameters from off-chip memory to the on-chip scratchpad. By the eighth cycle, the HT reads all malicious fields and waits for the target parameters to load, replacing them with the malicious values at the right moment.

Passing different malicious information to the accelerator will result in varying attack effects. The previous adversarial weight attack methods primarily aimed to flip the fewest bits to achieve either untargeted attacks (i.e., reducing model inference accuracy) (Rakin et al., 2019) or targeted attacks (e.g., N to 1 attack, single sample attack) (Rakin et al., 2021; Bai et al., 2023). However, these methods typically explore all layers of the model, including the first convolutional layer and the final classification layer, which are often the most sensitive and attract the attention of security engineers. As shown by the red dots in Fig. 3, the identified bits are usually distributed irregularly across the layers, which means that the HT would require significant hardware resources to locate them. This contradicts the design principle of HT, which should aim to use as few hardware resources as possible. Additionally, more malicious information would need to be em-

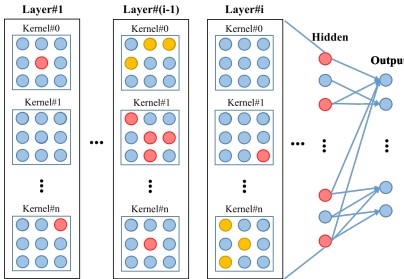

Figure 3: Two adversarial weight attack methods are shown. Blue dots represent clean parameters, red dots show scattered attack targets from method#1, and yellow dots from method #2 are clustered in the middle layers within a single kernel per layer.

bedded in the software layer (exceeding 15 fields, as detailed in Section 4.3), increasing the risk of detection by security engineers. Thus, we have outlined two challenges that need to be overcome in this threat model: **1)** Minimizing the number of target parameters and kernels across layers is preferable. Fig. 3 shows two attacking methods as an example. In contrast to the attack method indicated by red dots, the attack represented by yellow dots targets only one kernel in each of two convolutional layers, attacking three parameters per kernel, which is highly desirable from a hardware Trojan perspective. **2)** Avoid attacking the model's first layer and the final classification layer, as attacks on these two layers can have a significant impact on the model and are therefore more likely to be closely scrutinized by security engineers.

---

[1] The filter refers to the collection of kernels applied to a specific input feature map, while the kernel is a small matrix (e.g., $3 \times 3$) used for convolution to extract features

---

**Algorithm 1:** Pseudo-code of Cross-layer Sensitive Filter Exploration

**Input:** Model $model$, Layer $layer$, Test dataset $xTest$, Target Category $targetCatego$
1 , Exploration Algorithm $ExpA$
   **Output:** Parameter positions and corresponding bit flip information
2 **for** $f = 1...filterNum(layer)$ **do**
3     $iSet \leftarrow$ Randomly select a set of kernels from the filter $f$
4     **for** $i = 1...iSet$ **do**
5        $kSet \leftarrow$ For each $i$, use K-SIM to select the corresponding kernel from the prior layers
6        // Use $ExpA$ to attack $kSet$ with Equation 1 as fitness function.
7        $c_{idx}^{targetCatego} \leftarrow ExpA(model, kSet, xTest)$
8        **if** $c_{idx}^{targetCatego} > threshold$ **then**
9           $pSet \leftarrow$ Select the $x$ most negative parameters per kernel ($x$ set by the attacker)
10           // Re-explore $pSet$ using Equation 2 as the fitness function to identify the minimum number of bit flips.
11           $bSet \leftarrow PBPS(model, ExpA, pSet, xTest)$
12           $return\ bSet$
13        **end**
14     **end**
15 **end**

---

## 3.4 Cross-layer Sensitive Filter Exploration

The proposed C-SFE algorithm addresses the challenges outlined in Section 3.3. By focusing on the intermediate convolutional layers and avoiding both the initial convolutional and FC layers, C-SFE expands the decision boundary of a targeted category by flipping a few bits in one kernel per layer, enabling an N-to-1 attack. C-SFE is structured into two phases: The first phase, kernel-level exploration, primarily identifies sensitive kernel locations related to the specified category. The second phase, bit-level exploration, focuses on determining how to achieve the desired attack effect with the fewest bit flips, based on the selected kernels.

The core of kernel-level exploration is based on heuristic algorithms, such as genetic algorithms (GA) and PSO. The following example uses GA, assuming $N$ kernels of size $k \times k$ need to be explored simultaneously, and the GA with a population size of $PS$ generates $PS$ individuals per iteration, each comprising new $N \times k \times k$ parameters. These parameters sequentially replace the original values. After each replacement, inference is performed on all input images. The perturbation of the model by the new parameters is defined as follows:

$$c_{idx} = \sum_{i=1}^{I}(f(\boldsymbol{X}_i, \hat{\mathbf{W}}_{idx}) - f(\boldsymbol{X}_i, \mathbf{W})), \quad \text{where } idx \in [0, PS\text{-}1] \tag{1}$$

Where $f$ describes the behavior of the model, and its output is a vector consisting of confidence values for each category (i.e., the result after softmax). The $\boldsymbol{X}_i$ represents the input image, with a total of $I$ images. $\mathbf{W}$ represents the original parameters of the model, and $\hat{\mathbf{W}}_{idx}$ represents the $idx$-th individual in the population, differing from $\mathbf{W}$ in that $N \times k \times k$ parameters have been replaced with new values. $c_{idx}$ is a vector that stores the total confidence scores for each class, generated by the model after performing inference on all inputs using the $\hat{\mathbf{W}}_{idx}$. In the selection process of GA, individuals with high fitness are chosen and retained for the next generation. Assuming the goal is for the new kernel to bias the model towards classifying the input into the category $targetCatego$, the selection process will sort the values of the $c_{idx}^{targetCatego}$ from highest to lowest, and the corresponding $\hat{\mathbf{W}}$ of the top-ranked items will be retained for the next generation. It is worth noting that when $N = 1$, Equation 1 represents the exploration of a single kernel.

The kernels in the model define decision boundaries, with each contributing differently to each category. If explore kernel#1 and kernel#2 increases confidence for a category, joint exploration them will strongly bias the model towards that category. Otherwise, joint exploration will not bias the model. Therefore, selecting $N$ kernels at appropriate positions for joint exploration directly impacts the final attack effect. A naive method is to explore one kernel at a time, sequentially examining all kernels in the layer, and recording the category each kernel biases the model towards. Finally, for the specified category, select $N$ relevant kernels and conduct joint exploration on them, ultimately causing the model to classify most inputs into that category.

However, this method incurs high time complexity, especially when cross-layer attacks are implemented, as the time complexity scales with the number of layers involved. Given the necessity to conceal HT insertion during DSEs, it is critical to minimize the exploration time to avoid detection.

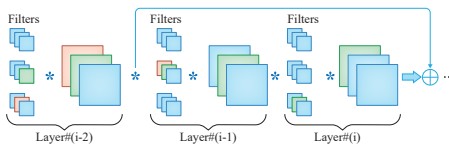

The proposed kernel-level exploration, combined with the Kernel Selection Inference Method (K-SIM), can directly infer the positions of relevant kernels in previous layers based on the kernel position in the current layer, thus eliminating the overhead of repeated exploration across multiple layers. The basic idea of K-SIM is to establish strong disturbance relationships within each layer, where only one kernel needs to be selected in each layer. Let $\mathbf{W}_l^{kIdx, fIdx}$ denote the kernel to be explored, where $\mathbf{W}$ represents the model parameters, $l$ is the layer number, and $fIdx$ and $kIdx$ denote the filter position in the layer and the kernel position within the filter, respectively. As shown in Fig. 4, the green parts indicate the method of selecting kernels for each layer in a sequential network structure (e.g., VGG). Suppose we need to explore $\mathbf{W}_i^{2,3}$, its corresponding input feature map is the second channel of the output feature map from Layer#(i-1), and the second filter in Layer#(i-1) can disturb this part. We can freely select $kIdx$, in this example, $\mathbf{W}_{i-1}^{2,2}$ is chosen. Using this kernel position, we can deduce in the same way that $\mathbf{W}_{i-2}^{free,2}$ should be selected for Layer#(i-2). The red parts illustrates the selection of the kernel in Layer#(i-2) when the model has a residual structure. In this scenario, to establish a direct perturbation relationship between Layer#(i-2) and Layer#i, the third filter must be selected in Layer#(i-2). Consequently, the kernel index in Layer#(i-1) is no longer freely selectable but must also be the third one.

Each kernel consists of multiple parameters (e.g., 9 in a $3 \times 3$ kernel). Not all parameters contribute equally to the model's bias towards a specified category, some are critical, while others have minimal impact. After kernel-level exploration identifies relevant kernels, bit-level exploration targets the critical parameters within them. We propose the Parameter Bit-Flip Priority Strategy (PBPS), which prioritizes the $x$ smallest values in the kernel, leaving the others unchanged, and then explores these $x$ values. If altering only these parameters can make $c^{targetCatego}$ reach the predefined threshold, then these parameters are the final targets for the attacker. For quantized models (e.g., int8), this method incorporates the Hamming distance between the original and target values of the quantized final targets as a penalty term into the fitness calculation of the GA, as shown in the following equation:

$$f_{idx} = c_{idx}^{targetCatego} - \beta \times HD(qNew, qOrg), \text{where } idx \in [0, PS\text{-}1] \qquad (2)$$

where $HD$ represents the Hamming distance, $qNew$ and $qOrg$ represent the modified and original quantized values, respectively, and $\beta$ is the penalty coefficient. Algorithm 1 details the process of C-SFE. Notably, the proposed C-SFE is one possible implementation that addresses the challenges outlined in Section 3.3. Considering the importance of concealing exploration time in automated accelerator generation, it is essential to develop more optimal algorithms.

## 4 EXPERIMENTAL RESULT

### 4.1 EXPERIMENTAL SETUP

The system follows Fig. 1, using the Rocket Core (Asanovic et al., 2016) as the controller and Gemmini $32 \times 32$ as the accelerator, with a hardware flow based on the vivado-risc-v project (Tarassov, 2024). We extended it to run at 90 MHz on the Xilinx U50 Alveo accelerator card. Linux runs on the Rocket Core, with ONNX Runtime 1.18.0 Python wheel compiled, embedding RoCC commands for Gemmini control. To validate the proposed attack, we use three model architectures: VGG-16 (Simonyan & Zisserman, 2014), ResNet-18 (He et al., 2016), and YOLOv8m-cls (Reis et al., 2023). Pre-trained models from the PyTorch model zoo are used for VGG-16 and ResNet-18, while Ultralytics' YOLOv8m-cls is used for YOLOv8. The dataset corresponding to these three models is ILSVRC 2012 (Deng et al., 2009). We randomly select 50 images from the validation set for C-SFE, with the rest used for verifying the algorithm's effectiveness. During actual execution, these models are quantized to int8, consistent with the quantization method provided by Gemmini.

### 4.2 ATTACKS RESULT

Table 1: Attack Performance on Different Models

| Model | Total Parameters | Target Category | Attack layers | Parameters per Kernel | Perturbation Rate (%) | Total flipped bits | Top 1/5 Clean (%) | Top 1/5 Malicious (%) | Targeted Classification Rate (%) |
|---|---|---|---|---|---|---|---|---|---|
| VGG-16 | 138,365,992 | n03884397 panpipe, pandean pipe, syrinx | Conv layers 5 to 8 | 7/7/7/7 | 0.0000202 | 85 | 71.1/90.2 | 0.2/1.2 | 98.1 |
| ResNet-18 | 11,689,512 | n03530642 honeycomb | layer2: (0): conv1 layer1: (1): conv2, conv1 | 4/4/4 | 0.000102 | 32 | 69.2/87.6 | 0.1/1 | 99.2 |
| YOLOv8m-cls | 17,053,336 | n03530642 honeycomb | model.4.m.3.cv1 model.4.m.2.cv2 model.4.m.2.cv1 | 3/3/3 | 0.0000527 | 24 | 75.3/92.7 | 0.21/1.4 | 97.3 |

The proposed attack method successfully targets any intermediate layer. Table 1 [2] shows examples for VGG-16, ResNet-18, and YOLOv8m-cls. For VGG-16, targeting *panpipe* and attacking the 5th to 8th convolutional layers, C-SFE identifies one kernel per layer, requiring attacks on seven parameters per kernel (85 bit flips in total), reducing Top-1 accuracy from 71.1% to 0.2% with only 0.0000202% parameter modification. Similarly, for ResNet-18 and YOLOv8m-cls, targeting *honeycomb*, attacking three intermediate layers with 12 and 9 parameter modifications drops the Top-1 accuracy to 0.1% and 0.21%, respectively.

Notably, the attack's effectiveness varies with the number of parameters attacked. Fig. 5 shows the minimum bit flips needed to achieve the highest classification rate for the *honeycomb* class in ResNet-18, targeting the same kernel positions as detailed in Table 1. Using the proposed C-SFE, it takes approximately 8 minutes to locate the kernels associated with this class. For instance, with 9 attacked parameters (3/3/3: one kernel per layer, three parameters per kernel), 95.8% of images are classified as *honeycomb* and the model's Top-1 and Top-5 accuracies drop to 0.21% and 1.5%, respectively, requiring 30 bit flips. For 8 parameters (3/3/2) and 7 parameters (3/2/2), the classification rate differs by 24.1%. Attacking 6 parameters (3/2/1) results in a 42.4% decrease in the classification rate compared to 7 parameters. Thus, attacking 7 parameters is the threshold for this model concerning this class. From the perspective of a single layer, the attack is concentrated on one kernel. For example, in YOLOv8m-cls, when the tile-based computation block containing the target kernel is about to start computing, the malicious information embedded in the control instructions only needs two fields for kernel's position and three fields for bit-flip mask information.

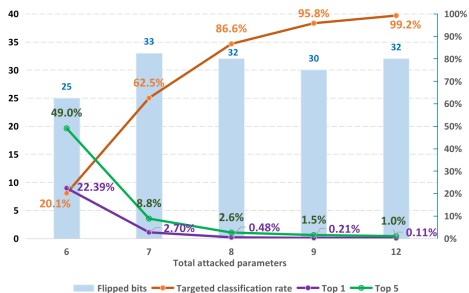

Figure 5: Impact of attacked parameters on 'honeycomb' classification rate and model accuracy in ResNet-18.

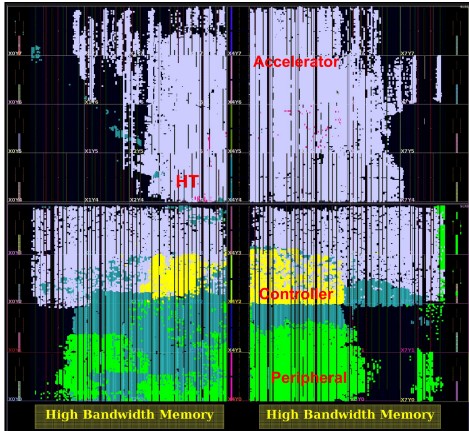

Figure 6: The layout of the design, with the HT hidden within the LoopConv module and the Scratchpad Controller of the accelerator.

Fig. 6 shows the hardware design layout: green sections represent peripherals (e.g., Ethernet controller, memory controller, UART), the yellow section is the single-core Rocket controller, and the gray-white section is the Gemmini-based accelerator. The HT is shown in red, divided into two parts: one in the convolution control unit to receive malicious information, and the other in the scratchpad control unit to attack parameter values at the right time, as illustrated in Fig. 2b. Table 2 shows the changes in hardware resources between the clean design without HT and the malicious design with

---

[2]More examples at https://github.com/AnonymousCode-HT/C-SFE.git

HT in the first four columns, where LUT usage increased by 0.34%, while other resources showed minimal changes.

Notably, during the synthesis process, different optimization strategies can be chosen. The last two columns of the Table 2 show that when the optimization strategy for the malicious design is area-focused, it uses fewer resources than the clean design. Since synthesis strategies are typically transparent to users, this aspect could potentially be exploited by attackers to make the HT more stealthy. Appendix A.2 presents the resource consumption of HT under different optimization modes.

Table 2: Resource Utilization Comparison Between Clean and Malicious Designs

| | Clean design | Malicious design | | | |
| | Strategy: Default | Strategy: Default | | Strategy: Area Opt | |
| | without HT | with HT | var. | with HT | var. |
|---|---|---|---|---|---|
| LUT | 332503 | 333642 | 0.34% | 319544 | -3.90% |
| LUTRAM | 30616 | 30598 | -0.06% | 30598 | -0.06% |
| FF | 314185 | 314227 | 0.01% | 314226 | 0.01% |
| BRAM | 452.5 | 452.5 | 0.00% | 452.5 | 0.00% |

### 4.3 IN COMPARISON WITH RELATED WORK

The usage of hardware resources for the HT depends not only on the number of bits flipped but also on the number of kernels and parameters targeted. Unlike previous bit-level adversarial weight attacks (Rakin et al., 2019; 2021; Bai et al., 2023), the proposed C-SFE focuses on flipping bits within a single kernel per layer, minimizing the required hardware resources. As an example, Fig. 7 shows the attack results of C-SFE and T-BFA (Rakin et al., 2021) on the 904th category of ILSVRC 2012 using ResNet-18. In this case, we restricted the layers explored by T-BFA to match those of C-SFE. Both methods successfully identified parameters that caused more than 98% of input images to be classified into this category, but T-BFA required attacking 11 kernels with uneven distribution, while C-SFE targeted only three kernels, one per layer. To transmit this information to the accelerator, T-BFA requires 23 fields for `layer1.1.conv1` (L1.1_C1 in Fig. 7), while C-SFE only needs 6 fields (detailed in Appendix A.3). As mentioned in Section 3.3, the fields used to transmit malicious information are valuable. For example, in Gemmini, if more than 15 fields are required, additional instructions must be sent to the accelerator, which could disrupt instruction timing and increase the risk of detection.

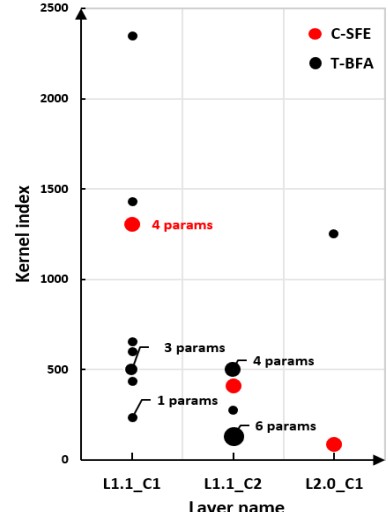

Figure 7: Comparison of target parameters identified by T-BFA and C-SFE on ResNet-18 for the 904th category of ILSVRC 2012. Both methods target the middle three convolutional layers (x-axis), with kernel positions on the y-axis, where different sizes represent the number of bits attacked in each kernel.

## 5 CONCLUSION

In this paper, we present a novel threat model for AI accelerator design paradigms, revealing potential security vulnerabilities. Specifically, we maliciously leverage the exploration unit to locate sensitive model parameters, embed hardware Trojans through the code generation module, and trigger them using malicious information hidden in communication instructions. We also highlight the challenges of implementing this threat model, as previous attack algorithms are not well-suited for it. To address these challenges, we propose a tailored attack algorithm, C-SFE. The effectiveness of the proposed threat model is validated on the Gemmini accelerator generation platform, and the attack performance was tested on three different models: VGG-16, ResNet-18, and YOLOv8m-cls. For each model, 4 kernels in VGG-16, 3 kernels in ResNet-18, and YOLOv8m-cls were attacked, with corresponding total parameter modifications of 28, 12, and 9, respectively. This resulted in over 98%, 99%, and 97% of the inputs being classified into the attacker's specified category for each model.

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

# A APPENDIX

## A.1 SIMPLIFIED ATTACK PROCESS

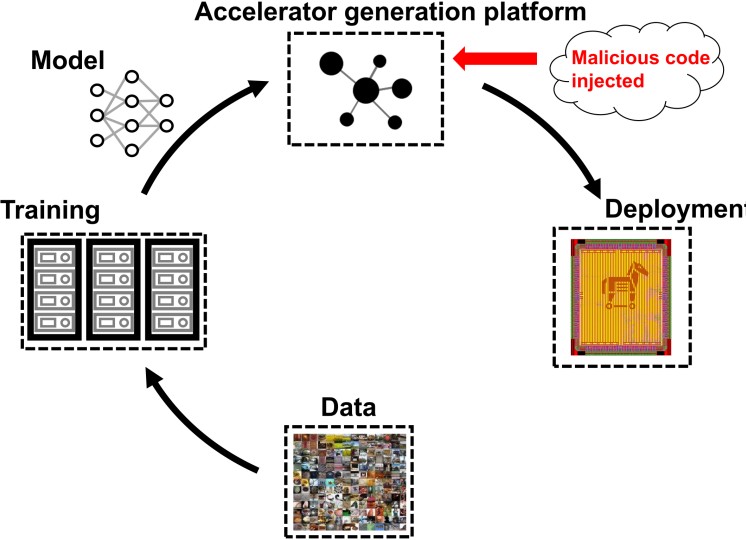

Figure 8: Simplified attack process

In the literature, numerous potential AI threats have been identified at various stages from data preparation and training to inference and deployment. These threats can deceive models, alter trained models, tamper with inference results, or extract sensitive details. Our work specifically targets the AI hardware generation phase, focusing on identifying potential threats within an automated AI accelerator generation platform by inserting HTs into the generated AI accelerators. The simplified attack process of our unique method is illustrated in Figure 8.

## A.2 HT SOURCE COMPARE

Table 3: Resource Utilization Comparison Between Clean and Malicious Designs

| | Clean design | | | Malicious design | | | | | |
|---|---|---|---|---|---|---|---|---|---|
| | Default | Area Opt | Performance Opt | Default | | Area Opt | | Performance Opt | |
| | without HT | without HT | without HT | with HT | var. | with HT | var. | With HT | var. |
| LUT | 332,503 | 318,295 | 332,364 | 333,642 | 0.34% | 319,544 | 0.39% | 333,582 | 0.37% |
| LUTRAM | 30,616 | 30,616 | 30,616 | 30,598 | -0.06% | 30,598 | -0.06% | 30,598 | -0.06% |
| FF | 314,185 | 314,185 | 314,178 | 314,227 | 0.01% | 314,226 | 0.01% | 314,258 | 0.03% |
| BRAM | 452.5 | 452.5 | 452.5 | 452.4 | 0.00% | 452.5 | 0.00% | 452.5 | 0.00% |

Table 3 compares the differences in resource consumption between clean designs and malicious designs under Default mode, Area Optimization mode, and Performance mode. The *Variation* represents the rate of change in resource usage between clean and malicious designs within the same mode.

## A.3 ATTACK METHOD COMPARISON

Table 4: Attack Method Comparison

|  | N-to-1 Attack | Top-1 Accuracy $\leq$0.1% | Bits Flipped | Num. Parameters | Num. Kernels | Num. Fields | HT Compatible (<15 Fields) |
|---|---|---|---|---|---|---|---|
| **Random Attack** | No | Yes | 356,877 | 100,000 | 33,664 | 18,936 | No |
| **T-BFA** | Yes | Yes | 21 | 21 | 11 | 23 | No |
| **C-SFE (Proposed)** | Yes | Yes | 28 | 12 | 3 | 6 | Yes |

Table 4 compares the performance of three attack methods: Random Attack, T-BFA, and C-SFE on the ResNet-18 model pretrained from the PyTorch model zoo. The dataset used was ILSVRC 2012. For Random Attack, we focused on the number of parameters that need to be modified to reduce the model's Top-1 accuracy to 0.1%. For T-BFA and C-SFE, corresponding to Fig. 7 (i.e., the parameter modifications cause over 98% of input images to be classified into the 904th category of ILSVRC 2012).

We employ a mainstream accelerator based on the systolic array architecture, in which the accelerator processes only one layer of the model at a time. Consequently, the resource consumption of HT and whether the ISA can accommodate malicious information depend solely on the layer with the highest number of attack parameters. For T-BFA, as illustrated in Fig. 7, we observe that attention should be focused on L1.1_C1 (a total of 9 attack parameters, distributed across 7 different kernels). In contrast, for C-SFE, since the attack parameters are uniformly distributed, there is no need to concentrate on a specific layer.

In Section 3.3, we indicate that attackers can leverage up to 15 fields during the instruction transmission process in Gemmini to insert attack information, including the positions and values of each parameter to be attacked. The calculation formula for fields is:

$$\text{Num. Fields} = \max_{l} \left( 2 \times \text{Num. Kernels}_l + \text{Num. Parameters}_l \right) \tag{3}$$

where $l$ represents the attacked layer. Num. Kernels$_l$ and Num. Parameters$_l$ represent the number of kernels and parameters attacked in the l-th layer, respectively (their sum corresponds to the Num. Kernels and Num. Parameters in Table 4). Each kernel is multiplied by 2 because it needs to pass the filter position to which the kernel belongs and its offset within that filter to the accelerator. Therefore, for the parameters identified by T-BFA in Fig. 7, the number of fields required for L1.1_C1 is $7*2+9=23$, whereas for C-SFE, the number of fields required is $1*2+4=6$. Therefore, since the parameters explored by T-BFA are unevenly distributed across layers and scattered within each layer, it cannot be considered an HT-friendly attack algorithm.

Table 5: Effectiveness of Defense Methods Against the Proposed Threat Model

| Defense Method | Detection Capability |
|---|---|
| Model-based methods (He et al., 2020) | ✗ |
| Machine learning-based RTL detection (Yu et al., 2021) | ✗ |
| Hash or checksum computation (Li et al., 2021; Liu et al., 2020a; Guo et al., 2021) | ✗ |
| Test vectors (Chakraborty et al., 2009; Saha et al., 2015) | ✗ |
| Information-Flow Tracking (Hu et al., 2016) | ✗ |

## A.4 DISCUSSION

We utilized the five commonly used defense methods introduced in Table 5 to evaluate the stealthiness of the proposed threat model. The results demonstrated that none of them could effectively defend against the proposed threat model.

**1)** Model-based methods enhance resistance to attacks by adjusting and optimizing the model itself. In our experiments, we retrained the model for 40 epochs using Weight Clustering (He et al., 2020) and then re-applied the proposed attack process. Experimental results show that achieving the same attack effectiveness on the retrained model requires targeting just one additional parameter per

layer. Therefore, while this approach slightly mitigates the attack's impact, it remains insufficient to effectively defend against the proposed threat model.

**2)** Machine learning-based detection of RTL is limited to a finite dataset. Taking the Trust-Hub dataset (Trust-Hub, 2024) as an example, HW2VEC (Yu et al., 2021) convert benchmarks into the form of Data-Flow Graph (DFG) or Abstract Syntax Tree (AST) representation, and then use Graph Convolutional Networks (GCN) to train data. The training dataset includes the PIC16F84 microprocessor and RS232 serial port. As proof, We input the malicious RTL code as test data into the model, the results show that the model cannot detect the maliciousness of the code.

**3)** Hash or checksum computation (Li et al., 2021), (Liu et al., 2020a), (Guo et al., 2021), which utilize hash or checksum values of off-chip memory (e.g., DRAM), have been proven effective in detecting attacks like BFA and T-BFA. These attacks specifically aim to alter bits in off-chip memory, therefore the use of hash functions makes it a reliable method for detecting DRAM-based attacks. However, the proposed HT resides within the accelerator's logic unit and targets only parameters stored in the on-chip memory, leaving those in off-chip memory unaffected. It is also worth noting that, the insert position of hardware Trojans is much more flexible making it challenging to determine the optimal location for inserting the hash computation unit. Consequently, the proposed threat model can evade detection by such verification methods.

**4)** Test vectors (Chakraborty et al., 2009), (Saha et al., 2015) are effective in detecting HTs located near the output. For example, in encryption chips, HTs are often deployed along key-related paths and positioned near the output. However, the proposed Payload design executes the attack by modifying internal model parameters without directly impacting the output. This greatly reduces the detection success rate of these methods, potentially rendering them incapable of identifying the attack.

**5)** Information-Flow Tracking (Hu et al., 2016) monitors the flow of data within a system to identify and prevent unauthorized data transmission. However, for CNNs, it is challenging to determine which parameters need to be tracked. Additionally, using this technique to track an excessive number of parameters can exponentially increase the hardware complexity of the accelerator. Therefore, it is more suitable for scenarios involving the protection of a single parameter, such as encryption keys in cryptographic chips.

