# OpenReview forum: "A Novel Security Threat Model for Automated AI Accelerator Generation Platforms"
_ICLR.cc/2025/Conference — Submitted to ICLR 2025_

### Official Review · Reviewer_vWaG · 2024-10-27

**Soundness:** 3
**Presentation:** 2
**Contribution:** 2
**Rating:** 5
**Confidence:** 3

**Summary:**

This paper presents a new threat model for AI accelerators. The proposed attack includes a complete pipeline that first finds out the sensitive parameters in the user-provided model and then embeds malicious code using bit flipping. The embedded Hardware Trojan (HT) is then triggered using information hidden in communication instructions. Evaluation results on three vision models show that the proposed attack can obtain a high attack success rate.

**Strengths:**

This paper has the following strengths:
+ The authors identify the vulnerability of AI accelerators in the context of a complete supply chain, starting from the pre-trained model from user to accelerator generation and then to deployment.
+ The authors decompose the problem of HT insertion into multiple sub-problems, i.e., locating the sensitive kernels, and bit-level perturbation. The attack is also designed with the goal of ensuring stealthiness and minimal hardware overhead.
+ Empirical results show that the proposed attack can achieve a high attack success rate with very few bit flips.

**Weaknesses:**

The paper has the following weaknesses:
- The preliminary section does not provide any information about the prior works on bit-flipping attacks. The lack of discussion makes it hard to understand the real innovation of this work since previous attacks have explored how to identify the sensitive kernels in the pre-trained quantized model (e.g., ProFlip).
- The proposed attack method is different from the previous model-level bit-flipping attacks since this paper is concerned with the hardware implementation of HT insertion. It's not clear why we cannot directly apply the existing bit-flipping attacks on AI accelerators. The challenges need to be clarified and empirical results can be used to support this.
- The design of Cross-layer Sensitive Filter Exploration (C-SFE) is not clear. The authors mentioned in Section 3.4 that 'CSFE expands the decision boundary of a targeted category by flipping a few bits in one kernel per layer'. However, the paper does not explain whether different layers are equally important and whether the attack can exploit a single layer only.

**Questions:**

Please consider addressing the weakness points above.

---

> ### Author Response · Authors · 2024-11-25
> **Response to reviewer vWaG**
>
> Thank you for your valuable suggestions! We have addressed the concerns you raised below.
>
> ***
> **W1. The preliminary section does not provide any information about the prior works on bit-flipping attacks. The lack of discussion makes it hard to understand the real innovation of this work since previous attacks have explored how to identify the sensitive kernels in the pre-trained quantized model (e.g., ProFlip).**
>
> Thank you for raising this important concern.
>
> In the paragraph starting at line 242 of the manuscript, we have discussed existing bit-flipping attack methods such as BFA and T-BFA, and provided actual comparison results in Section 4.3. In these paragraphs, we not only review the main objectives and strategies of existing bit-flipping attack methods but also thoroughly analyze their limitations, particularly emphasizing that prior works do not adapt well to HT designs. Building on this analysis, we present our innovation: the proposed C-SFE algorithm, which is better suited for HT design and offers enhanced stealthiness.
>
> We chose not to include detailed implementation specifics of prior works (such as mathematical formulas) in the preliminary section because our proposed C-SFE algorithm does not rely on these works. Instead, the proposed algorithm is a completely new attack scheme specifically tailored for HT, demonstrating independent innovation. Therefore, detailing the implementation aspects of prior works does not substantially aid in understanding our work and may instead make the content appear redundant.
>
> ***
> **W2. Why we cannot directly apply the existing bit-flipping attacks on AI accelerators. The challenges need to be clarified and empirical results can be used to support this.**
>
> Thank you for your insightful question.
>
> In the paragraph starting at line 242 of the manuscript, we explain why we cannot directly apply existing bit-flipping attacks on AI accelerators. This is because current methods adopt an attack model combining adversarial weight attacks with physical attacks on DRAM (e.g., T-BFA + Row Hammer attack). In these methods, the constraint is that the fewer bits of the model parameters are flipped, the better. However, our proposed threat model is based on adversarial weight attacks combined with a HT (i.e., C-SFE + HT). Compared to the previous attack model, this approach also needs to consider the hardware resource consumption of the hardware Trojan. Therefore, we need to add additional constraints on the regularity of the attack locations and the number of parameters in the adversarial weight attack (in the experimental part of the manuscript, the model uses 8-bit quantization, so each parameter consists of 8 bits).
> In Section 4.3, we compare the proposed attack method with existing methods. The results show that existing attack methods cannot be directly combined with hardware Trojans.
>
> ***
> **W3. The design of Cross-layer Sensitive Filter Exploration (C-SFE) is not clear. The authors mentioned in Section 3.4 that 'CSFE expands the decision boundary of a targeted category by flipping a few bits in one kernel per layer'. However, the paper does not explain whether different layers are equally important and whether the attack can exploit a single layer only.**
>
> Thank you for your comments. Our responses are provided below for each point.
>
> > **Whether different layers are equally important ?**
>
> During the development of our attack algorithm, we discovered that for N-to-1 attacks, different target classes require attacking different layers. Therefore, the significance of each layer depends on the target class.
> Furthermore, we would like to clarify that in the proposed algorithm, the number of kernels and parameters attacked for each layer are identical (a kernel consists of 3×3 parameters). This regular constraint is designed to accommodate the design of hardware Trojans.
>
> > **Whether the attack can exploit a single layer only ?**
>
> During the algorithm design process, we discovered that implementing an N-to-1 attack is achievable by targeting a single layer. However, compared to attacking multiple layers, it requires a larger number of parameters to be compromised. For example, using ResNet-18 as shown in Table 1 of the manuscript, a single-layer attack approximately requires compromising 40 parameters, whereas a cross-layer attack only needs to target 12 parameters (4 per layer). Since most AI accelerators operate using Systolic Array architectures, the hardware Trojan design for a single-layer attack would need hardware resources capable of locating and modifying these 40 parameters. In contrast, a cross-layer attack only requires hardware resources to locate and modify 4 parameters. Therefore, the latter approach utilizes fewer hardware resources. Considering these two points, we abandoned the single-layer attack scheme during the design process.
> ***

---

> ### Author Response · Authors · 2024-11-29
> **Official Comment by Authors**
>
> Dear Reviewer vWaG,
>
> As the discussion period is nearing its conclusion, we kindly ask if you could review our response to ensure it addresses your concerns. Your feedback is greatly appreciated.
>
> Thank you for your time!
>
> Best,
>
> Authors

---

### Official Review · Reviewer_y7Ko · 2024-10-30

**Soundness:** 2
**Presentation:** 1
**Contribution:** 2
**Rating:** 3
**Confidence:** 4

**Summary:**

This paper attacks an AI accelerator generation platform using a unique threat model and adopting a genetic algorithm. The attack is evaluated across benchmark DNN models.

**Strengths:**

The paper explores a unique direction in exploiting security threats in automated AI accelerator platforms. The research direction is exploratory in nature and innovative.

**Weaknesses:**

Overall, the paper is very hard to follow, and the threat needs to be more explicit and clarify the assumptions. Since this is a unique direction, the authors should try to establish the threat model first. The summary of the weakness:

1. The threat model is vague and unclear on some assumptions. For example, sections 3.1 and 3.2 are confusing regarding what is the goal of the attacker. Is it a hardware trojan attack or an adversarial weight attack? Both terms are used. However, the threat model assumption is not equivalent for both cases. Moreover, there is discussion on how the attacker would gain access to model parameters, but hardly any discussion on how they will modify the parameters. The only discussion in the paper states in line 194, but how an attacker would insert this malicious code.

2. The most concerning part is it is not clear whether the attacker can access the training stage or not. If not, then how can they modify model parameters? In the model production stage and inference stage where exactly the attacker is located.

3. Some of assumptions are also not well established and needs clarified. For example line 265 "Avoid attacking the model’s first layer an the final classification layer, as attacks on these two layers can have a significant impact on the model and are therefore more
 likely to be closely scrutinized by security engineers." I doubt if that is the principle machine learning engineers follow; even if that is the case, how do they scrutinize these layers? There is no standard list of model parameters.

4. Attack algorithm is naively adopting a genetic algorithm. Eqn. 1 is very similar to computing gradient, since the attacker can access model parameters why not compute the gradient. The authors claim their method does not need gradient computation, more justification is needed why GA would be faster than computing gradients.

5. Figure 7 is confusing; the description is not clear on the conclusion. Both attacks target the middle three convolution layers, but what does that mean from an attack efficacy and efficiency perspective?

6. The figure-2 needs to improve, the text are hard to read on part (b).

**Questions:**

Overall, I commend the authors for exploring a unique security problem. However, the paper in its current form requires a lot of improvement before being ready for publication. Please refer to the weakness section.

---

> ### Author Response · Authors · 2024-11-25
> **Response to reviewer y7Ko (1/3)**
>
> Thank you for your valuable suggestions! We have addressed the concerns you raised below.
> ***
> **W1. The threat model is vague and unclear on some assumptions. For example, sections 3.1 and 3.2 are confusing regarding what is the goal of the attacker. Is it a hardware trojan attack or an adversarial weight attack? Both terms are used. However, the threat model assumption is not equivalent for both cases. Moreover, there is discussion on how the attacker would gain access to model parameters, but hardly any discussion on how they will modify the parameters. The only discussion in the paper states in line 194, but how an attacker would insert this malicious code.**
>
> Thank you for your comments. Our responses are provided below for each point.
>
> > **Sections 3.1 and 3.2 are confusing regarding what is the goal of the attacker. Is it a hardware trojan attack or an adversarial weight attack? Both terms are used. However, the threat model assumption is not equivalent for both cases.**
>
> We believe that the terms "adversarial weight attack" and "hardware Trojan" do not conflict for the following reasons:
>
> 1. An adversarial weight attack primarily refers to an algorithm at the software level, aiming to identify sensitive bits or parameters within the model. When executing the attack, methods such as Row-Hammer Attack (RHA) are employed to flip data bits in DRAM. Most prior works have adopted such approaches (i.e., adversarial weight attack + RHA), as exemplified by BFA and T-BFA introduced in line 243 of the manuscript.
>
> 2. Our proposed threat model employs an adversarial weight attack combined with a hardware Trojan (HT) (i.e., adversarial weight attack + HT), presenting an innovative approach to uncover security vulnerabilities in automated AI accelerator generation platforms that have been widely studied but lack sufficient security focus. In the paragraph starting at line 184 of the manuscript, we provide a detailed explanation of how the adversarial weight attack collaborates with HT, which does not conflict with the threat model introduced in Section 3.1.
>
> > **There is discussion on how the attacker would gain access to model parameters, but hardly any discussion on how they will modify the parameters.**
>
> The paragraph associated with line 194 in the manuscript primarily introduces how the proposed threat model attacks designs that adhere to the AI accelerator design paradigm. This outlines a general attack process and, therefore, does not delve into implementation specifics.
> As a concrete implementation, we demonstrate this attack process using Gemmini as an example in Section 3.3. Additionally, in line 237, we provide a detailed explanation of how HT modifies the parameters in conjunction with Figure 2(b).
> Specifically, when the Trigger is activated, the Payload's task is simple address comparison and data overwriting.
>
> > **How an attacker would insert this malicious code ?**
>
> The hardware Trojan consists of fixed code embedded within the accelerator template. For the implementation of the HT, please refer to the Trigger design (see line 1196 in the `LoopConv.scala` provided in the anonymous GitHub link in the manuscript) and the Payload design (see line 544 in `Scratchpad.scala`).
> ***
>
> **W2. The most concerning part is it is not clear whether the attacker can access the training stage or not. If not, then how can they modify model parameters? In the model production stage and inference stage where exactly the attacker is located.**
>
> Thank you for your comments. Our responses are provided below for each point.
>
> > **Whether the attacker can access the training stage or not?**
>
> Our answer is **NO**. As illustrated in the manuscript (starting from line 145), we emphasized in the threat model that the information accessible to the attacker is limited to the model structure, model parameters, and a small set of images used for model quantization calibration. However, hyperparameters, optimizers, and training datasets used during the model training phase are not included. Therefore, the attacker cannot access the training stage.
>
> > **If not, how can attacker modify model parameters?**
>
> The functionality of the proposed C-SFE is not to further optimize the accuracy of the user-input model but rather to explore which parameters in the model are sensitive to specific classes. Therefore, even without access to information from the training phase (e.g., training dataset), C-SFE can still identify the model's sensitive parameters based solely on a small subset of the dataset used for quantization calibration. Additionally, AI accelerator generation platforms have the authority to modify the model; for instance, during the quantization phase, they need to read the model parameters, calculate the quantization scale, and rewrite it into the model. These aspects align with previous studies (e.g., BFA, T-BFA).
> ***

---

> > ### Comment · Reviewer_y7Ko · 2024-11-26
> > **Thanks for the response**
> >
> > Thanks to the author for the response. My concern regarding the threat model persists.
> >
> > 1. HT requires a set of assumption for trigger activation and payload generation, then RH would require several system level assumption and privileges, I agree they do not conflict but they make the overall threat model really strong and hardly realistic.
> >
> > 2. It is fine that the paper does not require training level privileged, But what I mean is you still need to compute the gradient same as BFA and T-BFA. How would you compute that without the privilege of model query and back-propagation.
> >
> > 3. System level RH attack implementation discussion is limited and requires further experiments to validate the attack on a real setting.

---

> > > ### Author Response · Authors · 2024-11-27
> > > **Official Response by Authors (1/2)**
> > >
> > > We sincerely appreciate your thoughtful feedback. Below, we provide our detailed responses to each point.
> > >
> > > ***
> > > > HT requires a set of assumption for trigger activation and payload generation, then RH would require several system level assumption and privileges, I agree they do not conflict but they make the overall threat model really strong and hardly realistic.
> > >
> > > We would like to clarify that **the proposed threat model does not involve Row-Hammer (RH)**.
> > >
> > > The table below clearly compares previous works with our own, corresponding to the first sub-question in W1. From the **Attack Mechanism** column in the table, it can be seen that previous methods have utilized RH when executing attacks, whereas the proposed method employs hardware Trojan (HT).
> > > We use HT instead of RH because HT can be concealed within the logic units of the accelerator during the generation process in automated AI accelerator design platforms. In contrast, RH targets off-chip memory (e.g., DRAM) and cannot be integrated into these platforms.
> > >
> > > Although most of these platforms are currently still in the research stage, we aim to use the proposed threat model to expose potential security vulnerabilities. This will enable subsequent researchers to further explore defensive methods based on this foundation, thereby better addressing security in future commercial applications. Therefore, this threat model holds practical significance.
> > > The proposed threat model assumes that the attacker is the designer of the platforms, which is a plausible scenario. Experimental results also confirm the feasibility of such attacks, further supporting the effectiveness of the threat model.
> > >
> > >
> > > | **Attack Method**             | **Exploration Algorithm** | **Attack Mechanism** | **Hardware Attack Location** |
> > > |-------------------------------|---------------------------|---------------------------|------------------------------|
> > > | Previous (e.g., BFA, T-BFA)   | Backpropagation          | Row-Hammer            | Off-chip Memory              |
> > > | Proposed                      | Forward Propagation       | Hardware Trojan       | Logic Units                  |
> > >
> > > ***

---

> > > ### Author Response · Authors · 2024-11-27
> > > **Official Response by Authors (2/2)**
> > >
> > > ***
> > > > It is fine that the paper does not require training level privileged, But what I mean is you still need to compute the gradient same as BFA and T-BFA. How would you compute that without the privilege of model query and back-propagation.
> > >
> > > We would like to clarify that **the proposed algorithm does not require computing the gradient**, and is therefore different from BFA and T-BFA.
> > > As illustrated in the **Exploration Algorithm** in the table above, it relies solely on forward propagation, as described at line 97 of our manuscript.
> > > A simplified example demonstrates this process:
> > >
> > > Suppose the user inputs a trained model that can classify images as either Dog or Cat. The proposed algorithm performs an N-to-1 attack targeting the Dog class.
> > > First, we input two images used for quantization calibration. Forward propagation yields the model's prediction scores, as shown below:
> > >
> > > | Input Image | Dog Confidence Score | Cat Confidence Score | Predicted Label | Actual Label |
> > > |-------------|-----------------------|-----------------------|-----------------|--------------|
> > > | #1          | 0.2                   | 0.8                   | Cat             | Cat          |
> > > | #2          | 0.3                   | 0.7                   | Cat             | Cat          |
> > >
> > > We calculate the initial Total Dog Score as follows:
> > >
> > > **Total Dog Score = 0.2 + 0.3 = 0.5**
> > >
> > > Next, the proposed algorithm perturbs the kernels of intermediate convolutional layers. Assuming two layers are attacked, with one kernel attacked per layer.
> > >
> > > **Attack kernel#100 in Layer#3**
> > >
> > > |  **0.1**  |    **0.15**     |  **-0.06**  |                      |  **-0.1**  |    **0.03**     |  **0.03**  |
> > > |-----------|----------------|-----------|----------------------|-----------|----------------|-----------|
> > > |  **0.06**  |    **0.08**     |  **-0.21**  |          ⟶           |  **0.2**  |    **-0.11**     |  **0.06**  |
> > > |  **0.05**  |    **-0.1**     |  **0.05**  |                      |  **-0.1**  |    **0.01**     |  **0.26**  |
> > >
> > > **Attack kernel#200 in Layer#4**
> > >
> > > |  **0.05**  |    **0.12**     |  **-0.08**  |                      |  **0.1**  |    **-0.03**     |  **0.2**  |
> > > |-----------|----------------|-----------|----------------------|-----------|----------------|-----------|
> > > |  **-0.24**  |    **0.22**     |  **-0.04**  |          ⟶           |  **0.19**  |    **-0.13**     |  **0.01**  |
> > > |  **0.16**  |    **-0.04**     |  **0.02**  |                      |  **-0.1**  |    **0.08**     |  **0.09**  |
> > >
> > > Notably, the new values remain within the range of post-training quantization (PTQ).
> > >
> > > After applying these perturbations, forward propagation is performed again. The new prediction results are shown below:
> > >
> > > | Input Image | Dog Confidence Score | Cat Confidence Score | Predicted Label | Actual Label |
> > > |-------------|-----------------------|-----------------------|-----------------|--------------|
> > > | #1          | **0.7**                   | 0.3                   | **Dog**             | Cat          |
> > > | #2          | 0.4                   | 0.6                   | Cat             | Cat          |
> > >
> > > The updated Total Dog Score is calculated as:
> > >
> > > **Total Dog Score = 0.7 + 0.4 = 1.1**
> > >
> > > The Total Dog Score Difference, defined as the increase in the score, is:
> > >
> > > **Total Dog Score Difference = 1.1 - 0.5 = 0.6**
> > >
> > > The significant increase in the **Total Dog Score** shows that perturbing these kernels causes the model to favor the Dog class, misclassifying images like the first image from Cat to Dog. A genetic algorithm (GA) is then used to iteratively optimize the kernel values based on the **Total Dog Score Difference** as the fitness function, without relying on gradients. This approach maximizes the score difference, enhancing the attack's effectiveness.
> > >
> > > The above steps correspond to lines 6-7 in Algorithm 1 of the manuscript, where the **Total Dog Score Difference** corresponds to $c_{idx}^{targetCatego}$. Kernel selection for each layer is detailed in the paragraph starting at line 330, which introduces the Kernel Selection Inference Method (K-SIM).
> > >
> > > This algorithm adopts an HT-friendly approach by systematically targeting one kernel per layer, aligning with the architecture of mainstream Systolic array-based accelerators, which process one layer at a time. Unlike backpropagation-based methods, which often lead to uneven kernel distribution across layers, this method ensures consistent hardware resource allocation for attacking each layer. Details are discussed in the paragraph starting at line 242 of the manuscript.
> > >
> > > ***
> > > > System level RH attack implementation discussion is limited and requires further experiments to validate the attack on a real setting.
> > >
> > > As mentioned in the response to the first question, we did not use the RH attack. Our threat model employs only HT, and in the experimental section, we applied it to the current mainstream Gemmini-based accelerator. Its effectiveness was validated using three models: YOLOv8m-cls, ResNet-18, and VGG-16.
> > > ***

---

> ### Author Response · Authors · 2024-11-25
> **Response to reviewer y7Ko (2/3)**
>
> ***
> **W3. Some of assumptions are also not well established and needs clarified. For example line 265 "Avoid attacking the model’s first layer an the final classification layer, as attacks on these two layers can have a significant impact on the model and are therefore more likely to be closely scrutinized by security engineers." I doubt if that is the principle machine learning engineers follow; even if that is the case, how do they scrutinize these layers? There is no standard list of model parameters.**
>
> In our work, the reasons for us to avoid attacks on the model's first and final classification layers are based on the following considerations:
>
> 1. Importance of Key Layers: The first layer is usually responsible for extracting primary features from the input data, while the final classification layer directly affects the model's output results. Tampering with the parameters of these two layers can lead to a significant decline in model performance. For example, TBT [1] and SBA [2] exploit this characteristic. Additionally, the parameters identified by T-BFA also tend to be distributed in the final classification layer.
>
> 2. Priority in Security Detection: In actual security reviews and model validation processes, security engineers often prioritize parts that have a significant impact on overall performance. Due to the special status of the first and last layers, they are more likely to become focal points of inspection.
>
> Regarding your questions about "how to scrutinize these layers" and "there is no standard list of model parameters," our understanding is:
>
> 1. Parameter Verification: Although there is no unified list of parameters, security engineers may perform checks on the model's key parameters (especially layers that have a significant impact) [3], [4].
>
> 2. Anomaly Behavior Monitoring: Security engineers may detect anomalies by monitoring the model's input and output behaviors. For example, if the output distribution of the model has significantly changed, they may trace back to inspect the output layer and its parameters.
>
> [1] Adnan Siraj Rakin, Zhezhi He, and Deliang Fan. Tbt: Targeted neural network attack with bit trojan. In Proceedings of the IEEE/CVF Conference on Computer Vision and Pattern Recognition, pp.13198–13207, 2020.
>
> [2] Yannan Liu, Lingxiao Wei, Bo Luo, and Qiang Xu. Fault injection attack on deep neural network. In 2017 IEEE/ACM International Conference on Computer-Aided Design (ICCAD), pp. 131–138. IEEE, 2017.
>
> [3] Qi Liu, Wujie Wen, and Yanzhi Wang. Concurrent weight encoding-based detection for bit-flip attack on neural network accelerators. In IEEE/ACM International Conference on Computer-Aided Design (ICCAD), 2020, 2020a.
>
> [4] Jingtao Li, Adnan Siraj Rakin, Zhezhi He, Deliang Fan, and Chaitali Chakrabarti. Radar: Run-time adversarial weight attack detection and accuracy recovery. In 2021 Design, Automation & Test in Europe Conference & Exhibition (DATE), pp. 790–795. IEEE, 2021.
> ***

---

> ### Author Response · Authors · 2024-11-25
> **Response to reviewer y7Ko (3/3)**
>
> ***
> **W4. Attack algorithm is naively adopting a genetic algorithm. Eqn. 1 is very similar to computing gradient, since the attacker can access model parameters why not compute the gradient. The authors claim their method does not need gradient computation, more justification is needed why GA would be faster than computing gradients.**
>
> Thank you for your comments. Our responses are provided below for each point.
>
> > **Eqn. 1 is very similar to computing gradient.**
>
> **We believe that Equation 1 is fundamentally different from computing gradients.**
> Equation 1 calculates the cumulative difference in the model's output confidences between the original parameters and the modified parameters across all inputs. This method assesses the overall impact of parameter changes on the model's predictions. In contrast, computing gradients involves calculating derivatives of a loss function with respect to the model parameters, which is used for optimization during training. Since our focus is on evaluating the changes in output distributions rather than optimizing the parameters, computing gradients is not applicable in this context.
>
> > **Why not compute the gradient ?**
>
> In the paragraph starting at line 242 of the manuscript, we explain why we cannot directly apply existing gradient-based attack methods on AI accelerators. This is because current gradient-based attack methods adopt an attack model combining adversarial weight attacks with physical attacks on DRAM (e.g., T-BFA + Row Hammer attack). In these methods, the constraint is that the fewer bits of the model parameters are flipped, the better. However, our proposed threat model is based on adversarial weight attacks combined with a hardware Trojan (i.e., C-SFE + HT). Compared to the previous attack model, this approach also needs to consider the hardware resource consumption of the hardware Trojan. Therefore, we need to add additional constraints on the regularity of the attack locations and the number of parameters in the adversarial weight attack (in the experimental part of the manuscript, the model uses 8-bit quantization, so each parameter consists of 8 bits).
>
> > **More justification is needed why GA would be faster than computing gradients.**
>
> We would like to clarify that **in our manuscript, we did not claim that the Genetic Algorithm is faster than computing gradients**.
>
> The original intention of this manuscript is to reveal the lack of security research related to the AI accelerator design paradigm by proposing a threat model. Since there has been no prior research combining adversarial weight attacks and hardware Trojans, as mentioned in line 360 of the manuscript, our proposed C-SFE algorithm is merely a feasible solution that satisfies this threat model; it cannot guarantee optimality but can serve as a baseline for future related research.
>
> ***
> **W5. Figure 7 is confusing; the description is not clear on the conclusion. Both attacks target the middle three convolution layers, but what does that mean from an attack efficacy and efficiency perspective?**
>
> The primary aim of Section 4.3, which corresponds to Figure 7, is to highlight the suitability of our proposed C-SFE method for hardware Trojan implementation (as stated in the first sentence of Section 4.3), particularly in contrast to previous attack algorithms like T-BFA.
>
> To address the confusion regarding Figure 7:
>
> 1. Attack Efficacy: In line 457, we emphasized that both attacks achieve similar misclassification rates, demonstrating that C-SFE does not compromise on effectiveness despite targeting fewer kernels.
>
> 2. Attack Efficiency: From line 460, we emphasized that C-SFE is more efficient in terms of hardware resource utilization and information transmission, making it better suited for integration with hardware Trojans.
>
> ***
>
> **W6. The Figure 2 needs to improve, the text are hard to read on part (b).**
>
> Thank you for your suggestion. Figure 2 has been revised for readability improvement.
> ***

---

### Official Review · Reviewer_UkBM · 2024-11-03

**Soundness:** 2
**Presentation:** 2
**Contribution:** 2
**Rating:** 3
**Confidence:** 4

**Summary:**

In this paper, the authors propose a threat model in which the accelerator provider acts as an attacker by injecting hardware Trojans (HTs) through an EDA tool into the accelerator to enable misclassification attacks in Deep Neural Network (DNN) inference. They categorize these attacks into three types: 1-to-1, N-to-1, and others.

**Strengths:**

The paper assumes the EDA tool developer of the AI accelerator platform as the adversary, which is a compelling area of research.

**Weaknesses:**

The paper assumes that the EDA tool developer of the AI accelerator platform acts as the adversary, which is a compelling area of research. However, it is essential to demonstrate that the added HTs are challenging to detect by verification tools. In reality, substantial work has been done on HT detection and hardware formal verification. That said, the paper does not discuss whether existing HT detectors [1-5],  could identify the injected Trojans, including such an analysis would enhance the credibility of the claims.

[1] Bhunia, Swarup, Michael S. Hsiao, Mainak Banga, and Seetharam Narasimhan. "Hardware Trojan attacks: Threat analysis and countermeasures." Proceedings of the IEEE 102, no. 8 (2014): 1229-1247.

[2] Gubbi, Kevin Immanuel, Banafsheh Saber Latibari, Anirudh Srikanth, Tyler Sheaves, Sayed Arash Beheshti-Shirazi, Sai Manoj PD, Satareh Rafatirad, Avesta Sasan, Houman Homayoun, and Soheil Salehi. "Hardware trojan detection using machine learning: A tutorial." ACM Transactions on Embedded Computing Systems 22, no. 3 (2023): 1-26.

[3] Chakraborty, Rajat Subhra, Francis Wolff, Somnath Paul, Christos Papachristou, and Swarup Bhunia. "MERO: A statistical approach for hardware Trojan detection." In International Workshop on Cryptographic Hardware and Embedded Systems, pp. 396-410. Berlin, Heidelberg: Springer Berlin Heidelberg, 2009.

[4] Lyu, Yangdi, and Prabhat Mishra. "Scalable activation of rare triggers in hardware trojans by repeated maximal clique sampling." IEEE Transactions on Computer-Aided Design of Integrated Circuits and Systems 40, no. 7 (2020): 1287-1300.

[5] Saha, Sayandeep, Rajat Subhra Chakraborty, Srinivasa Shashank Nuthakki, Anshul, and Debdeep Mukhopadhyay. "Improved test pattern generation for hardware trojan detection using genetic algorithm and boolean satisfiability." In Cryptographic Hardware and Embedded Systems--CHES 2015: 17th International Workshop, Saint-Malo, France, September 13-16, 2015, Proceedings 17, pp. 577-596. Springer Berlin Heidelberg, 2015.

Designing ASICs follows strict standards. SkyNet and similar projects rely on HLS rather than ASIC processes, which may facilitate deployment on FPGA but does not reflect the final circuit. For automatically inserted Trojans, can they pass through functional verification steps?

For IoT applications, it would be more appropriate to use platforms like the Ultra96 (as used in SkyNet) or KRIA KR260, which are MPSoC-based devices commonly utilized in IoT contexts.

It appears that the LoopConv module was mapped to LUTs, yet Fig. 6 does not clearly show the specific size and location. Based on Table 2, using the default optimization requires approximately 1,000 additional LUTs. This amount seems quite large for HT injection.

The authors inject a Trojan into the EDA tool, potentially enabling triggers or malicious optimization on the hardware. I believe this point should be emphasized, with more detail on the significance and application value of such EDA tools. However, this brings up a broader issue: an EDA tool superior to others would likely dominate the market, making it challenging for a provider to practically act as an attacker.

**Questions:**

Could the authors explain why they chose the U50 platform (a high-end, expensive database acceleration card) despite targeting IoT applications?

For inserted Trojans, can they pass through functional verification steps?

Could you provide component consumption under a fair comparison scheme, such as comparing a clean accelerator vs an HT-injected accelerator, both using Vivado's Area Optimization mode, Default mode, latency optimization, and so on?

Could the authors discuss how realistic their threat model is?

---

> ### Author Response · Authors · 2024-11-25
> **Response to reviewer UkBM (1/3)**
>
> Thank you for your valuable suggestions! We have addressed the concerns you raised below.
> ***
> **Q1. Could the authors explain why they chose the U50 platform ? For IoT applications, it would be more appropriate to use platforms like the Ultra96 or KRIA KR260.**
>
> The selection of U50 was primarily driven by its ability to support a larger systolic array (32×32) Gemmini-generated accelerator, compared to the 16 × 16 array on the Xilinx Kintex-7 series (such as xc7k325t). This larger array significantly enhances inference speed, which is critical for efficiently testing the accuracy of quantized models during our experiments, as illustrated in Figure 1 of the manuscript. On the other hand, we also acknowledge that platforms like the Ultra96 or KRIA KR260, equipped with ARM Cortex-A53 processors, may be more typical for IoT applications due to their balance of computational power and energy efficiency. However, our threat model is designed to be hardware-agnostic and applicable to various configurations, including those with smaller computational resources like the ARM Cortex-M series.
>
> The tests on U50 and the Xilinx Kintex-7 (xc7k325t) board were conducted to demonstrate the flexibility and adaptability of our attack framework rather than suggest a specific IoT implementation. The Vivado project files for the Kintex-7 are available via the anonymous GitHub link provided in the manuscript.
>
> In summary, while the U50 platform was chosen for its specific technical advantages in our experimental setup, the broader applicability of our threat model to IoT contexts remains valid across both high performance and low-power devices, as we've also validated on FPGA platforms similar to those you mentioned.
>
> ***
> **Q2. For inserted Trojans, can they pass through functional verification steps ?**
>
> Our response is: **Yes**. The reasons are as follows:
>
> 1. The Trigger module in the HT ensures its stealthiness. When the Trigger is not activated, the HT does not introduce any functional anomalies to the design. Since the HT's trigger conditions are difficult to satisfy, the Trojan cannot be detected during the regular function verification process.
>
> 2. In the proposed HT design, the primary responsibility of the Payload component is to replace target model parameters stored in on-chip memory with malicious values. Notably, this component's design does not directly affect the circuit's output, which represents a significant difference from previous Payload designs. For example, in encryption chips, HTs are typically deployed along key-related paths and positioned near the output. This arrangement means that the Payload's behavior directly impacts the chip's output, enabling security personnel to develop detection strategies based on this relationship. However, the proposed Payload design carries out the attack by replacing internal model parameters without directly affecting the output. This makes it difficult to detect using methods based on test vectors [1], [2] and information-flow tracking [3].
>
> 3. We have experimentally confirmed that machine learning-based detection methods [4] are also unable to detect the proposed HT. Additionally, using Weight Clustering [5] does not mitigate the proposed attack.
>
>
> [1] Chakraborty, Rajat Subhra, Francis Wolff, Somnath Paul, Christos Papachristou, and Swarup Bhunia. "MERO: A statistical approach for hardware Trojan detection." In International Workshop on Cryptographic Hardware and Embedded Systems, pp. 396-410. Berlin, Heidelberg: Springer Berlin Heidelberg, 2009.
>
> [2] Saha, Sayandeep, Rajat Subhra Chakraborty, Srinivasa Shashank Nuthakki, Anshul, and Debdeep Mukhopadhyay. "Improved test pattern generation for hardware trojan detection using genetic algorithm and boolean satisfiability." In Cryptographic Hardware and Embedded Systems--CHES 2015: 17th International Workshop, Saint-Malo, France, September 13-16, 2015, Proceedings 17, pp. 577-596. Springer Berlin Heidelberg, 2015.
>
> [3] Wei Hu, Baolei Mao, Jason Oberg, and Ryan Kastner. Detecting hardware trojans with gate-level information-flow tracking. Computer, 49(8):44–52, 2016.
>
> [4] Shih-Yuan Yu, Rozhin Yasaei, Qingrong Zhou, Tommy Nguyen, and Mohammad Abdullah Al Faruque. Hw2vec: A graph learning tool for automating hardware security. In 2021 IEEE International Symposium on Hardware Oriented Security and Trust (HOST), pp. 13–23. IEEE, 2021.
>
> [5] Zhezhi He, Adnan Siraj Rakin, Jingtao Li, Chaitali Chakrabarti, and Deliang Fan. Defending and harnessing the bit-flip based adversarial weight attack. In Proceedings of the IEEE/CVF Conference on Computer Vision and Pattern Recognition, pp. 14095–14103, 2020.
> ***

---

> ### Author Response · Authors · 2024-11-25
> **Response to reviewer UkBM (2/3)**
>
> ***
> **Q3. Could you provide component consumption under a fair comparison scheme, such as comparing a clean accelerator vs an HT-injected accelerator, both using Vivado's Area Optimization mode, Default mode, latency optimization, and so on? It appears that the LoopConv module was mapped to LUTs, yet Fig. 6 does not clearly show the specific size and location. Based on Table 2, using the default optimization requires approximately 1,000 additional LUTs. This amount seems quite large for HT injection.**
>
> Thank you for your insightful question.
>
> The table below presents a fair comparison scheme, comparing the differences in resource consumption between clean designs and malicious designs under Default mode, Area Optimization mode, and Performance mode. The Variation represents the rate of change in resource usage between clean and malicious designs within the same mode.
>
> We would like to clarify that, as described in the paragraph starting at line 435 of the manuscript, the original intention of additionally including the Area Optimization mode in Table 2 was to illustrate that users (software engineers) typically do not pay attention to the optimization options of the EDA tool. Therefore, this aspect could potentially be exploited by attackers to make the HT more stealthy.
>
>
> | **Resource** | **CD-Default** | **CD-Area Opt** | **CD-Perf Opt** | **MD-Default** | **Var.** | **MD-Area Opt** | **Var.** | **MD-Perf Opt** | **Var.** |
> |--------------|----------------|-----------------|-----------------|----------------|----------|-----------------|----------|-----------------|----------|
> |              | *without HT*   | *without HT*    | *without HT*    | *with HT*      |          | *with HT*       |          | *with HT*       |          |
> | **LUT**      | 332503         | 318295          | 332364          | 333642         | 0.34%    | 319544          | 0.39%    | 333582          | 0.37%    |
> | **LUTRAM**   | 30616          | 30616           | 30616           | 30598          | -0.06%   | 30598           | -0.06%   | 30598           | -0.06%   |
> | **FF**       | 314185         | 314185          | 314178          | 314227         | 0.01%    | 314226          | 0.01%    | 314258          | 0.03%    |
> | **BRAM**     | 452.5          | 452.5           | 452.5           | 452.4          | 0.00%    | 452.5           | 0.00%    | 452.5           | 0.00%    |
>
> LoopConv is used to parse Rocket Custom Coprocessor (RoCC) commands and convert them into a series of memory access and computation instructions (implementation details are provided in `LoopConv.scala` in the GitHub link in the manuscript). It is implemented as sequential logic based on a state machine.
> As part of the accelerator, LoopConv is located within the gray-white area in Figure 6. Since the main purpose of Figure 6 is to show the layout of various modules in the entire SoC (e.g., controller, peripherals, accelerator), we did not further subdivide the internal modules of the accelerator or label them with different colors. Instead, we highlighted the HT, as it is the most critical component, to emphasize its importance.
>
> We acknowledge that our HT implementation consumes about 1,000 additional LUTs under default optimization, which is indeed significant. This is partly because we used Chisel to write the HT code. While Chisel offers high-level hardware description capabilities, it may not leverage certain low-level optimization techniques available in Verilog, leading to extra resource usage.
> To reduce LUT consumption, the attacker could write the HT code directly in Verilog. In our threat model, this is feasible since the attacker has deep hardware design knowledge and can employ low-level optimizations. Directly using Verilog allows for finer control over the circuit and more efficient resource utilization, significantly reducing the additional LUT overhead.
> Therefore, although the current HT implementation requires slightly more resource usage, it does not affect the practicality and effectiveness of the proposed threat model.
>
> ***

---

> ### Author Response · Authors · 2024-11-25
> **Response to reviewer UkBM (3/3)**
>
> ***
> **Q4. The authors inject a Trojan into the EDA tool, potentially enabling triggers or malicious optimization on the hardware. I believe this point should be emphasized, with more detail on the significance and application value of such EDA tools. However, this brings up a broader issue: an EDA tool superior to others would likely dominate the market, making it challenging for a provider to practically act as an attacker? Could the authors discuss how realistic their threat model is?**
>
> We would like to clarify that the proposed HT is not embedded within any EDA tool itself, rather it is integrated into the Verilog files as indicated in lines 181 and 190 of the manuscript. Specifically, the HT is introduced in the source code and can be synthesized by EDA tools to generate the bitfile or layout, in which the EDA tool is merely a part of the VLSI design flow (as shown in Figure 1 of the manuscript), thus not compromising the EDA tool directly. As an example, the source code of HT can be found at line 1196 in `LoopConv.scala` and line 544 in `Scratchpad.scala` within the anonymous GitHub link provided in the manuscript. The Verilog file generated for synthesis with the EDA tool has also been uploaded to GitHub. Additionally, while a single EDA tool dominating the market might suggest a lower likelihood of it acting maliciously, security vulnerabilities remain an inherent risk. We believe dominance does not guarantee immunity from internal threats or supply chain compromises. For example, the security vulnerability in Intel HID Event Filter (CVE-2024-25561) [1] and Intel Agilex® FPGA Firmware Advisory (CVE-2024-25576) [2] demonstrate that even well-established tools can host vulnerabilities that allow escalated privileges or other security breaches.
>
> It is believed that the proposed threat model is realistic and should not be omitted. As mentioned in the paragraph starting from line 114 of the manuscript, our threat model focuses on the growing field of automated AI accelerator generation platforms, however it is a domain where security has not been extensively explored. By abstracting critical security modules as shown in Figure 1, we aim to expose potential security risks and ensure that it has adequate security guarantees when commercialized in the future.  In addition, our model has been validated across different hardware platforms and applies to any design framework adhering to the AI accelerator design paradigm, including but not limited to the Gemmini and GeneSys [3], [4] platforms. This widespread applicability underlines the practical relevance and urgency of addressing such threats.
>
> [1] Intel. Intel Security Advisory SA-01089, 2024. URL https://www.intel.com/content/www/us/en/security-center/advisory/intel-sa-01089.html. Accessed on November 14, 2024.
>
> [2] Intel. Intel Security Advisory SA-01087, 2024. URL https://www.intel.com/content/www/us/en/security-center/advisory/intel-sa-01087.html. Accessed on November 14, 2024.
>
> [3] Soroush Ghodrati, Sean Kinzer, Hanyang Xu, Rohan Mahapatra, Yoonsung Kim, Byung Hoon Ahn,Dong Kai Wang, Lavanya Karthikeyan, Amir Yazdanbakhsh, Jongse Park, et al. Tandem processor: Grappling with emerging operators in neural networks. In Proceedings of the 29th ACM International Conference on Architectural Support for Programming Languages and Operating Systems, Volume 2, pp. 1165–1182, 2024.
>
> [4] actlab genesys. Genesys, 2024. URL https://github.com/actlab-genesys/GeneSys. Accessed on November 14, 2024.
>
> ***

---

> ### Author Response · Authors · 2024-11-29
> **Official Comment by Authors**
>
> Dear Reviewer UkBM,
>
> As the discussion period is nearing its conclusion, we kindly ask if you could review our response to ensure it addresses your concerns. Your feedback is greatly appreciated.
>
> Thank you for your time!
>
> Best,
>
> Authors

---

### Official Review · Reviewer_8b4e · 2024-11-03

**Soundness:** 3
**Presentation:** 2
**Contribution:** 2
**Rating:** 5
**Confidence:** 4

**Summary:**

This paper presents a novel threat model targeting automated AI accelerator generation platforms. It says that full-stack integration in the design process of accelerators has become common. In this work, it explores how manipulating the model parameters will result in misclassification introducing a method called Cross-layer Sensitive Filter Exploration (C-SFE). To achieve this, in this threat model the attacker has to be active in three components of the platform; design space exploration (DSE), software stack, and hardware generation. In DSE, the attacker finds the parameters to be targeted. In hardware generation, HT is inserted to change the targeted parameters. In the software stack, there is an interface injection to trigger the HT. The results show high misclassification rates for YOLOv8, ResNet-18, and VGG-16. The area overhead of the HT is 0.34% of the total design.

**Strengths:**

1. The authors say that the security aspect of accelerator generation platforms is largely unexplored, and this is one of the initial works done to explore security vulnerabilities.

2. The proposed HT introduces a very low area overhead (0.34%) making it hard to detect before runtime.

3. The paper presents multiple evaluations to justify their claims; attack performance, # of parameters vs classification rate, layout design with HT, resource utilization, etc.

**Weaknesses:**

1. The evaluations of this threat model are limited to Gemmini. This can be further extended to other platforms such as GeneSys.

2. Although the area overhead of the accelerator is very low, it would have been better to have runtime performance numbers compared to the benign platform to see whether there is a significant difference in the inference pipeline. Further, would there be a larger difference when the test dataset size increases?

3. The discussion section (4.4) of the paper lacks depth. Perhaps to justify the findings of this section, it would be better to include them in the appendix.

4. The paper mentions minimizing the exploration time during DSE. However, no clear numbers were presented how this would change for being and malicious platforms.

5. In some written sections, the use of the word "parameter" becomes a bit confusing whether it is the model parameters of DSE parameters (ex: line 190).

6. Minor issue: double quotations were not properly handled. (line: 394)

7. Y axises of Figure 6 are not named.

**Questions:**

1. Would a simple validation of model parameters at the end of execution, along with the classification results, show that the parameters have changed? This is considering that given a trained model, we are aware of its model parameters.

2. Instead of carefully selecting which parameters to change, how would the results look like if parameters were changed randomly? I think this should be a baseline to be compared with. It is important to see whether a complex technique is required.

3. Why different numbers of kernels and parameters were selected for various models? Keeping them consistent across different models would be a good comparison to do.

---

> ### Author Response · Authors · 2024-11-25
> **Response to reviewer 8b4e (1/3)**
>
> Thank you for your valuable suggestions! We have addressed the concerns you raised below.
> ***
> **Q1. Would a simple validation of model parameters at the end of execution, along with the classification results, show that the parameters have changed?**
>
> Thank you for your insightful question.
>
> Our response is: **No**. The reasons are as follows:
>
> 1. The Trigger design in the HT ensures its stealth, as shown in line 197 of the manuscript. The random activation of the Trigger makes it difficult for security engineers to determine when to perform parameter validation.
>
> 2. When the Trigger is deactivated, the malicious parameters in the on-chip memory will be overwritten by clean model parameters from the off-chip memory at the next load, allowing the model to function normally.
>
> ***
> **Q2. Instead of carefully selecting which parameters to change, how would the results look like if parameters were changed randomly?**
>
> Thank you for bringing up this important aspect.
>
> The table below compares the number of parameters altered by the proposed C-SFE versus random attack and their impact on accuracy across three models. For random attack, each parameter count was tested in 10 trials, and the averages are reported. Taking ResNet-18 as an example, reducing the model's Top-1 accuracy to 0.1% requires modifying over 100,000 parameters with a random attack, whereas C-SFE achieves the same result by modifying only 12 parameters. The code for the random attack has been uploaded to the anonymous GitHub link provided in the manuscript.
>
>
> | **Attack Method** | **No. of Parameters** | **YOLOv8m-cls** |           | **ResNet-18** |           | **VGG-16** |           |
> |-------------------|-----------------------|-----------------|-----------|---------------|-----------|------------|-----------|
> |                   |                       | **Top 1(%)**    | **Top 5(%)** | **Top 1(%)**  | **Top 5(%)** | **Top 1(%)** | **Top 5(%)** |
> | **Clean**         | --                    | 76.6            | 93.3      | 69.8          | 89.1      | 71.6       | 90.4      |
> | **Random**        | 10                    | 76.6            | 93.3      | 69.7          | 89.1      | 71.6       | 90.4      |
> | **Random**        | 100                   | 76.4            | 93.2      | 69.7          | 89.1      | 71.6       | 90.4      |
> | **Random**        | 1,000                 | 76.9            | 92.8      | 68.9          | 88.6      | 71.6       | 90.4      |
> | **Random**        | 10,000                | 48.6            | 73.6      | 55.2          | 78.9      | 71.5       | 90.3      |
> | **Random**        | 100,000               | 0.1             | 0.6       | 0.1           | 0.8       | 69.9       | 89.4      |
> | **Propose**       | 28                    | --              | --        | --            | --        | **0.2**    | **1.2**   |
> | **Propose**       | 12                    | --              | --        | **0.1**       | **1.0**   | --         | --        |
> | **Propose**       | 9                     | **0.21**        | **1.4**   | --            | --        | --         | --        |
>
> It is noteworthy that exploration algorithms suitable for HT need to consider not only whether the attack parameters can achieve the desired effect but also whether locating these parameters during the accelerator's runtime phase will consume excessive hardware resources. Therefore, even if a random attack is effective, the random positioning of these parameters may cause HT to expend a significant amount of hardware resources in locating them. Regarding existing adversarial weight attacks, many studies outperform our proposed C-SFE algorithm in terms of the number of bits attacked (e.g., T-BFA). However, current adversarial weight attack methods do not impose constraints on the regularity of the identified parameter locations, making them unsuitable for HT design. This is the motivation behind our emphasis in the paragraph starting at line 242 of the manuscript on the need to design an HT-friendly attack algorithm.
> ***
>
> **Q3. Why different numbers of kernels and parameters were selected for various models?**
>
> Thank you for your insightful question.
>
> The selection of different numbers of kernels and parameters across various models is dictated by their varying capabilities in feature extraction. Our approach, C-SFE, adapts to these differences by selecting the set of target parameters for each model. It should also be noted that, in our threat model, the attacker's access is restricted to only 50 images used for quantization calibration. Our experimental results demonstrate that C-SFE can efficiently identify the minimal set of regular model parameters that classifies all 50 images into the target category. However, given the limited size of the available calibration dataset, which indicates the attacker does not have information about the full dataset, targeting additional parameters does not expand the attack’s impact beyond these 50 images.
> ***

---

> ### Author Response · Authors · 2024-11-25
> **Response to reviewer 8b4e (2/3)**
>
> ***
> **W1. The evaluations of this threat model can be further extended to other platforms such as GeneSys.**
>
> Thank you for bringing up this important aspect.
>
> As shown in line 206 of the manuscript, the proposed threat model can be applied not only to the Gemmini accelerator generation platform but also to GeneSys [1]. This is because GeneSys aligns with the AI accelerator design paradigm introduced in Figure 1 of the manuscript. Specifically:
>
> 1. "Cross-domain Applications" [2] corresponds to the user-input trained model (the blue section on the left in Figure 1).
> 2. "Architecture Stack" is used to determine the optimal hardware configuration (such as the number of processing elements, bit-widths, and on-chip buffer configurations) and corresponds to the "Exploration unit" in the Hardware build flow (the left yellow section in Figure 1).
> 3. "Compilation Stack" is used to convert the input model into Instruction Set Architecture (ISA). For the Tandem Processor, it generates Tandem ISA and Systolic Array ISA; for Gemmini, it generates RISC-V ISA and Rocket Custom Coprocessor (RoCC) commands. This part corresponds to the "Software build flow" (the right yellow section in Figure 1).
> 4. "FPGA/ASIC Implementation" corresponds to the "VLSI design flow" (the left yellow section in Figure 1).
>
> Therefore, in the GeneSys-based automatic accelerator generation platform, the proposed threat model can still explore the model's vulnerabilities during the "Cross-domain Applications" and "Compilation Stack" stages, and insert malicious information into the Tandem Processor's Systolic Array ISA.
>
> [1] Soroush Ghodrati, Sean Kinzer, Hanyang Xu, Rohan Mahapatra, Yoonsung Kim, Byung Hoon Ahn,Dong Kai Wang, Lavanya Karthikeyan, Amir Yazdanbakhsh, Jongse Park, et al. Tandem processor: Grappling with emerging operators in neural networks. In Proceedings of the 29th ACM International Conference on Architectural Support for Programming Languages and Operating Systems, Volume 2, pp. 1165–1182, 2024.
>
> [2] actlab genesys. Genesys, 2024. URL https://github.com/actlab-genesys/GeneSys. Accessed on November 14, 2024.
> ***
>
> **W2.**
>
> > **It would have been better to have runtime performance numbers compared to the benign platform.**
>
> Thank you for your insightful feedback.
>
> Our design ensures that HT does not affect inference time for the following reasons:
> - HT Trigger in sequential logic: As depicted in Figure 2(b) and detailed in the LoopConv.scala script (line 1196) available in the supplementary GitHub repository, the HT trigger can be designed in a sequential logic, ensuring non-intrusive execution.
> - Parallel execution: The Trigger and the Convolution configuration module in the accelerator execute in parallel, leading to overlapping execution times without incurring additional delays.
> - Payload in combinational logic: The payload of the HT, implemented as a simple combinational logic circuit (refer to Scratchpad.scala, line 544), meets timing constraints without causing timing violations.
> Furthermore, empirical tests conducted under the hardware configuration described in our manuscript show that performing inference on a single image using ResNet-18 architecture requires approximately 4,287,063 clock cycles, which demonstrates no significant deviation from the expected performance norms. Therefore, the presence of HT does not adversely affect the accelerator's runtime performance.
>
> > **Would there be a larger difference when the test dataset size increases?**
>
> Actually, our response is **NO**. There would not be a significant difference. Due to the deliberate design of the HT trigger, which is configured to activate very infrequently. Even as the dataset size increases, the probability of triggering the HT remains very low. This design ensures that the HT does not significantly impact the overall behavior of the system using different test dataset sizes, while maintaining a consistent and predictable performance.
> ***
>
> **W3. The discussion section (4.4) of the paper lacks depth. Perhaps to justify the findings of this section, it would be better to includethem in the appendix.**
>
> Thank you for your suggestion. We have moved it to the appendix.
>
> ***
> **W4. No clear numbers were presented how this would change for being and malicious platforms.**
>
> Thank you for highlighting this aspect.
>
> The exploration algorithm is executed solely during the generation phase of the accelerator (see the yellow section in Figure 1) and is not involved during the runtime phase. Specifically, during the generation phase, line 408 in the manuscript indicates that the runtime of C-SFE is approximately 8 minutes. It is noteworthy that the total time from the user's model input to bitfile generation is around 50 minutes. Since C-SFE runs in parallel with the overall process, it does not affect the total time.
> ***

---

> ### Author Response · Authors · 2024-11-25
> **Response to reviewer 8b4e (3/3)**
>
> ***
> **W5. In some written sections, the use of the word "parameter" becomes a bit confusing whether it is the model parameters of DSE parameters (ex: line 190).**
>
> Thank you for your comment. We have clarified them in the manuscript.
>
> ***
> **W6. Minor issue: double quotations were not properly handled. (line: 394)**
>
> Thank you for your comment. We have revised them in the manuscript.
>
> ***
> **W7. Y axises of Figure 6 are not named.**
>
> Figure 6 presents the Vivado-generated layout of the accelerator with HT, which is primarily used to illustrate the distribution and connectivity of internal resources within the FPGA. As this layout corresponds to a specific FPGA board, it does not feature labeled x-axis and y-axis. The HBM label below the image simply indicates that the accelerator utilizes High Bandwidth Memory (HBM) as its off-chip memory.
> ***

---

> > ### Comment · Reviewer_8b4e · 2024-11-27
> >
> > Thank you for your detailed response and clarifications. I appreciate your effort to address my concerns. I believe the additional experiments (especially, the random baseline) will add more depth to the paper making it stronger. However, my concerns regarding the robustness of the threat model remain. Following up on Q1, the use of a hash function for continuous and stateful validation of model parameters can counter the HT. By periodically computing and verifying hash values of the parameters throughout execution, any unauthorized modification can be detected regardless of when it occurs as we consider model parameters as an input. This eliminates the need to identify specific intervals for validation. This makes the robustness of the threat model questionable. Therefore, I will maintain my original scores.

---

> > > ### Author Response · Authors · 2024-11-28
> > > **Official Responses by Authors**
> > >
> > > Thank you for your valuable feedback and for taking the time to elaborate on your concerns.
> > > However, regarding your suggestion that using hash values of the parameters can detect any changes to the model parameters, we would like to clarify the following points:
> > >
> > > > **1. The use of a Hash function is an effective detection method for DRAM-based attacks.**
> > >
> > > Several methods such as [1], [2], [3] that utilize hash or checksum values of off-chip memory (e.g., DRAM) have been proven effective in detecting attacks like BFA and T-BFA. As detailed in the **Attack Mechanism** and **Hardware Attack Location** sections of the table below, BFA and T-BFA-based attacks aim to alter bits in off-chip memory, therefore the use of hash functions makes it a reliable method for detecting DRAM-based attacks.
> > >
> > > > **2. The use of Hash functions for model parameters is not suitable for detecting the proposed HT-based attacks. One reason is that due to the flexibility in HT insertion, potential HTs can be inserted in the logic unit after DMA.**
> > >
> > > Unlike the DRAM-based attacks described above, our proposed threat model is based on a hardware Trojan (HT) concealed within the logic units of an accelerator. Taking the Gemmini-based accelerator used in our work as an example, the corresponding memory hierarchy can be described as follows (Please also refer to Figure 2 in the manuscript for more details).
> > >
> > > **off-chip memory → L2 cache (CPU) → DMA (accelerator) → HT (accelerator)  → on-chip scratchpad (accelerator) → compute unit (accelerator)**
> > >
> > > It can be observed that the HT is inserted in the logic unit after the DMA. Therefore, even if a hash function unit is implemented for off-chip memory or cache, the inserted HT cannot be detected. It is also worth noting that, unlike Row-Hammer attacks which focus on off-chip memory, the insert position of HTs is much more flexible. For instance, they can be located in the **on-chip scratchpad (accelerator)** or in the **compute unit (accelerator)**. Despite this flexibility in HT insertion, we only demonstrate one general approach in the manuscript to illustrate the potential risk of the proposed threat model.
> > >
> > > > **3. Another reason why hash functions are not a feasible method against the proposed attack is the rare activation of the inserted HTs. Since the payload of these HTs only activates when triggered, the output from the accelerator remains correct whenever the HT is not triggered.**
> > >
> > > As described in our manuscript (Line 108), a HT usually consists of a trigger and a payload. During normal operations, the trigger will monitor a certain (or some) signal. When these signals reach a specific condition, it will control the payload into working state, the payload is responsible for the specific attack. Therefore, the payload of the inserted HT only works (i.e., the specific parameters are modified, and the output result goes wrong) when the HT is triggered, otherwise the output of the accelerator will be the same as that of a normal design. Because the HT trigger condition can be deliberately designed, the existing memory protection-based approaches are not a feasible method against the proposed attack.
> > >
> > > > **4. Summary**
> > >
> > > In summary, hash computations are effective for detecting Row-Hammer attacks targeting off-chip memory, but they cannot effectively detect attacks that employ HTs. This is why our answer to Q1 is **NO**.
> > >
> > > Since automated AI accelerator design platform security has not received sufficient attention, there are currently no effective defense mechanisms for the proposed threat model (i.e., adversary weight attack + HT). This research aims to inspire future researchers to focus on the reliability and security of these platforms.
> > >
> > > In Appendix A.4 of our manuscript, five defense techniques are used to comprehensively evaluate the stealthiness of the proposed threat model.
> > >
> > > | **Attack Method**             | **Attack Mechanism**     | **Hardware Attack Location** |
> > > |-------------------------------|--------------------------|------------------------------|
> > > | Previous (e.g., BFA, T-BFA)   | Row-Hammer              | Off-chip Memory              |
> > > | Proposed                      | Hardware Trojan         | Logic Units                  |
> > >
> > >
> > > [1] Jingtao Li, Adnan Siraj Rakin, Zhezhi He, Deliang Fan, and Chaitali Chakrabarti. Radar: Run-time adversarial weight attack detection and accuracy recovery. In 2021 Design, Automation & Test in Europe Conference & Exhibition (DATE), pp. 790–795. IEEE, 2021.
> > >
> > > [2] Qi Liu, Wujie Wen, and Yanzhi Wang. Concurrent weight encoding-based detection for bit-flip attack on neural network accelerators. In IEEE/ACM International Conference on Computer-Aided Design (ICCAD), 2020, 2020a.
> > >
> > > [3] Yanan Guo, Liang Liu, Yueqiang Cheng, Youtao Zhang, and Jun Yang. Modelshield: A generic and portable framework extension for defending bit-flip based adversarial weight attacks. In 2021 IEEE 39th International Conference on Computer Design (ICCD), pp. 559–562. IEEE, 2021.

---

### Author Response · Authors · 2024-11-25
**General Response**

We extend our heartfelt thanks to all the reviewers for their valuable insights and constructive feedback, which have been instrumental in enhancing and refining our approach. In the following responses, we address each reviewer’s individual questions and concerns. Additionally, we provide a summary of the revisions made to the manuscript.

## Summary of Paper Revisions
- Enhanced Figure 2 by enlarging the text in part (b) to improve its readability.
- Refactored line 190 to avoid confusion with the term "parameter."
- Revised Figure 7 by replacing "bits" with "parameters," correcting an error in our manuscript.
- Added an Appendix that includes:
  1. A simplified attack process (A.1).
  2. A comparison of hardware resource usage between designs with hardware Trojans (HT) and clean designs under various optimization strategies (A.2).
  3. An explanation of why existing adversarial weight attack methods are unsuitable for HT designs (A.3).
  4. A comprehensive evaluation of the proposed threat model's stealthiness (A.4).
- In the anonymous GitHub link, test code for random attacks has been added, along with TCL files for building projects on Xilinx Kintex-7 series chips and Verilog files containing malicious code.

---

### Meta-Review · Area_Chair_kbHP · 2024-12-22

**Metareview:**

The paper proposes a new threat model targeting automated AI accelerator generation platforms. The proposed attack includes a pipeline that first finds out the sensitive parameters in the targeted model provided by the user and then embeds malicious code using bit flipping. The embedded Hardware Trojan (HT) is then triggered using information hidden in the communication instructions.
All the reviewers agree that the proposed threat model is novel and interesting. However, several reviewers are concerned about the clarity of the paper, especially on the description of the threat model, including the ability of the attacker and the practicality of some assumptions. Reviewers are also skeptical about whether the inserted trojan can bypass the detection of existing verification tools.

**Additional Comments On Reviewer Discussion:**

Reviewers are skeptical about whether the inserted trojans can bypass the detection of existing verification tools. The authors provide responses, including claiming the inserted trojans cannot be detected by five detection methods in the appendix. However, the provided discussion is insufficient and does not include newer methods, such as [1]. The reviewers are not convinced that the proposed attack is robust enough.

[1] Puschner, Endres, et al. "Red team vs. blue team: a real-world hardware Trojan detection case study across four modern CMOS technology generations." 2023 IEEE Symposium on Security and Privacy (SP). IEEE, 2023.

---

### Decision · Program_Chairs · 2025-01-22

Reject